# Effect of sex/gender on obesity traits in Canadian first year university students: The GENEiUS study

**Tanmay Sharma**[1], **Rita E. Morassut**[1], **Christine Langlois**[1], **David Meyre** [1,2]*

**1** Department of Health Research Methods, Evidence and Impact, McMaster University, Hamilton, Canada,
**2** Department of Pathology and Molecular Medicine, McMaster University, Hamilton, Canada

* meyred@mcmaster.ca

## Abstract

### Background

While weight gain during first year of university has been well documented in North America, literature on sex-specific effects is scarce and inconsistent. The objective of this investigation was to explore sex-specific changes in obesity traits during first year of university at McMaster University (Ontario, Canada).

### Methods

245 first-year students (80.4% females) were followed longitudinally with data collected early in the academic year and towards the end of the year. Obesity parameters including weight, waist and hip circumferences, BMI, and waist to hip ratio were investigated. The Mann-Whitney U test and the Wilcoxon signed-rank test were used for pairwise comparison of traits in the absence of adjustments. Additionally, the repeated-measures ANOVA test was used with covariate adjustments to investigate the interaction between sex and time.

### Results

Overall sample trends indicated a significant increase in mean weight by 1.55 kg (95% CI: 1.24–1.86) over the school year (p<0.001). This was accompanied by significant gains in BMI, and waist and hip circumferences (p<0.001) in the overall sample. At baseline, males presented with higher body weight, BMI, waist and hip circumferences, and WHR, as compared to their females counterparts (p<0.01). Additionally, sex-stratified analysis indicated significant gains in weight, BMI, and waist and hip circumferences in both males and females (p<0.01). However, a comparison of the magnitude of change over time between the two sex groups revealed no significant difference for any of the investigated traits (p>0.05).

### Conclusion

While our study confirms significant weight gain in both male and female first year university students in Ontario, Canada, it does not show sex specific differences within this context.

**Data Availability Statement:** All relevant data are within the manuscript and its Supporting Information files.

**Funding:** DM holds a Canada Research Chair in Genetics of Obesity. TS is supported by the

Canadian Institutes of Health Research Canada Graduate Scholarship. The funders had no role in study design, data collection and analysis, decision to publish, or preparation of the manuscript.

Our investigation highlights the importance of accounting for sex and gender in health research and supports the need of further studies in this area.

## Introduction

According to recent World Health Organization (WHO) estimates, the global prevalence of obesity and overweight has nearly tripled over the last four decades with over one-third of the global population now having an estimated body mass index (BMI) above 25 kg/m$^2$. Canada is one of several high-income countries that has a high prevalence of obesity and overweight [1]. Data from the 2018 Canadian Community Health Survey (CCHS) indicate that approximately 63.1% of Canadians are either overweight or obese, with a higher proportion of males (69.4%) being affected than females (56.7%) on average. Interestingly, when examining the CCHS estimates of obesity/overweight prevalence in Canadians between 2010 and 2018, a considerable increase from approximately 23.24% to 31.2% is noted amongst individuals aged 18–19. This is highly concerning as previous studies have implicated elevated BMI during adolescence and young adulthood as an important risk factor for chronic obesity and other secondary complications later in life including higher morbidity and early mortality [2–5]. Despite the availability of different therapeutic interventions, ranging from diet adjustments to bariatric surgery, treatment of obesity remains a biomedical and public health challenge due to its multifactorial pathogenesis and, as such, the disorder tends to persist as a chronic condition in most affected individuals [6,7]. In that context, further research in this area for a better understanding of the disorder is critical to optimize the prediction, prevention, and treatment of obesity [8].

The period between the age of 17 and 25 years, sometimes referred to as "young adulthood", encompasses important transition events for many young adults, one of which includes starting post-secondary education [9]. Interestingly, while education status is negatively correlated with BMI in the general population from high-income countries, students pursuing post-secondary education have been shown to be at greater risk for weight gain than those not pursuing university education in the United States [10–13]. The "Freshman 15" is a popular belief that undergraduate university students gain 15 pounds (6.8 kg) during their first year of university studies [10,11]. While previous studies have supported the theory of weight gain during first year of university, the amount gained has been reported to be more modest at approximately 3 to 5 lbs (1.4 to 2.3 kg) [14–16]. This increase can be partially attributed to changes in environmental exposures and lifestyle habits, such as unhealthy diet, decreased physical activity, and increased sedentary behavior, which are usually observed during transition to university [17–20]. However, not everyone exposed to this 'obesogenic' environment becomes obese [21]. Several biological factors such as *in utero* programming, gut microbiome, epigenetics and genetics, can modulate an individual's susceptibility to weight gain and can help explain a portion of the observed inter-individual anthropometric variance [21,22]. Sex/gender (hereafter referred to as sex) represents an important risk factor at the interface of biology and environment, comprising of a set of biological and sociological constructs [23,24].

Previous research examining obesity traits in post-secondary students indicates that males present with a higher BMI than their female counterparts on average [25,26]. However, the literature on the effect of sex on weight gain during the academic year is mixed [25,27,28]. Canadian studies within this context have been relatively limited and have also reported contradictory results [29–32]. While some reports have indicated sex-specific anthropometric

change in first year Canadian university students, others have found no significant differences [29–32]. Most recently, Beaudry et al. reported a significant sex effect amongst first year students at a university in Ontario, showing that male students gain about twice as much weight as their female counterparts [29]. This has important implications as this report implicates sex as an important risk factor to be taken into consideration for prevention efforts [29]. Since we recruited first year students at a different university campus in Ontario, we carried out a follow-up investigation and attempted to replicate this observation in a cohort of 245 undergraduate students at McMaster University.

## Methods

### Study design and participants

Genetic and EnviroNmental Effects on weight in University Students (GENEiUS) is a prospective observational study which investigates the environmental and biological determinants of obesity trait changes in Canadian undergraduate students [8]. Undergraduate students from McMaster University (Hamilton, Ontario) are followed every six months over four years beginning in September of their first year of study. First year undergraduate students were primarily recruited via in-class advertising on main university campus and through social media promotion. First year students enrolled at McMaster University between the ages of 17 and 25 are eligible to participate in the study. Individuals who are pregnant, have given birth, or have a medical condition which can impact BMI for a long period of time (e.g. bariatric surgery, immobilization from injury) have been excluded from the study. Additional details regarding the GENEiUS study have been described previously [8]. Written informed consent was obtained directly from the participants. All methods and procedures for this study were in accordance with the Declaration of Helsinki principles and were reviewed and approved by the Hamilton Integrated Research Ethics Board (REB#0524).

### Data collection

Four cohorts of participants (2015–2016, 2016–2017, 2017–2018, 2018–2019) were followed longitudinally with data collected at two study visits: the beginning of their first-year (September/October) and towards the end of their first-year (March/April). A total of 361 participants were enrolled in the study. Two-hundred forty-five of them completed the baseline and follow-up visits and represent the analyzed sample in this report (N = 245; 80.4% females: 19.6% males). Data analyzed in this study included anthropometrics (weight, BMI, waist and hip circumference, waist to hip ratio), and demographics (sex, ethnicity, living arrangement, program of study). Anthropometric traits, including weight, height, waist circumference (WC), hip circumference (HC), were measured at baseline (September/October) and again at 6 months post-baseline (March/April). Weight was measured to the nearest 0.1 kg using a digital scale (Seca, Hamburg, Germany) and height was measured to the nearest 0.1 cm using a portable stadiometer (Seca 225, Hamburg, Germany). WC was measured at the midpoint of the last palpable rib and the superior portion of the iliac crest to the nearest 0.1 cm and HC was measured at the widest part of the buttocks to the nearest 0.1 cm using a stretch-resistant tape measure, in accordance with WHO guidelines. All anthropometric measurements were performed by trained research assistants. Additional obesity trait outcomes, including BMI and waist to hip ratio (WHR), were calculated. BMI (kg/m2) was calculated by dividing weight by squared height and WHR was calculated by dividing WC by HC. Information about demographic characteristics was collected at the first appointment using an online, self-reported questionnaire.

## Statistical analysis

All statistical analyses were performed using IBM SPSS Version 25 statistical package. Descriptive analysis was carried out to assess the preliminary distribution of traits within the study sample. Data for continuous variables have been reported using means and SD while categorical data have been reported by counts and percentages. Anthropometric data at each time point were screened for potential outliers. Any identified outlying data points were individually cross-checked to determine if they were true outliers, representing participants who truly fell outside the general distribution of our data, or if the outliers were a result of inaccuracies in measurement or data transcription. Data inaccuracies were corrected while all other outliers were left in the dataset. All data were assessed graphically and statistically for normality of distribution prior to analysis. The Mann-Whitney U test was used for a pairwise comparison of outcomes at baseline between males and females, in absence of adjustments for covariates. The Wilcoxon signed-rank test was used for a pairwise comparison of change in obesity traits from the beginning to the end of first year university, in absence of adjustments for covariates. The effect of sex on anthropometric change was assessed using a repeated measures analysis of variance (RMANOVA) [33]. An inverse normal rank transformation was applied to the anthropometric data for each time point, as previously reported [34,35]. Transformation resulted in the normality of the distribution for the anthropometric data. Different covariates including cohort of recruitment (i.e. 2015–2016, 2016–2017, 2017–2018, 2018–2019), faculty of study (i.e. science vs. non-science), and living arrangement (i.e. living in residence on campus, living at home, living in student housing off campus) were tested separately in each RMANOVA model. In this case, we followed the covariate adjustment strategy used by Beaudry *et al*. for the available traits [29]. As such, the covariates were only retained if their interaction with time was significant or marginally significant ($p<0.1$), otherwise reduced models were presented. Partial eta-squared values ($\eta^2$) from the RMANOVA were also presented as a measure of effect size [36]. Based on the fact that i) the present study is hypothesis-driven; ii) the research questions have been previously tested in literature; iii) tested obesity outcomes are not independent; applying a Bonferroni correction reduces the chance of making type I errors, but increases the chance of making type II errors [37,38]. Therefore, the level of statistical significance was set at $p<0.05$ for all tests.

## Results

### Participant characteristics

A total of 361 participants were enrolled into the study between 2015 and 2018 of which 245 (68%) completed one year of follow up (i.e. completed the first baseline visit around September/October and a second follow-up visit in March/April) between 2016 and 2019. The 245 participants that completed one year of follow up represent the analyzed sample in this report. The mean length of time between the baseline and follow-up visits was 21.6 weeks (SD = 2.18). Participants displayed an average age of 17.87 years (SD = 0.48) and female participants accounted for 80.4% of the analyzed sample (n = 197). Thirty one percent of the participants were East Asian (n = 76), 24.9% were white Caucasian (n = 61), 18.8% were South Asian (n = 46), 12.7% were mixed (n = 31), 6.9% were Middle Eastern (n = 17), and 5.7% (n = 14) collectively belonged to other ethnicity groups including African, Latin American, and Pacific Islander. In terms of living arrangement, 69.4% percent of the sample reported living in university residence on campus (n = 170), 19.6% reported living at home with family (n = 48), and 10.6% reported living in a student house off campus (n = 26). Among those who reported their program of study, 86.2% reported being enrolled in a science based academic program

(e.g. Health Science, Life Science, Kinesiology, Engineering) while 13.8% reported being in enrolled a non-science academic program (e.g. Humanities, Business, Arts).

## Anthropometric patterns in first year of university: Overall trends

Early on in the academic year (i.e. at baseline), the average body weight, BMI, WC, HC, and WHR for the overall sample was 60.42 kg (SD = 11.98), 21.52 kg/m2 (SD = 3.34), 75.08 cm (SD = 8.69), 97.18 cm (SD = 7.73), and 0.772 (SD = 0.050) respectively. Approximately 78.4% (n = 192) of the participants had a normal BMI, 12.2% were underweight (n = 30), 6.5% were overweight (n = 16), and 2.9% (n = 7) were obese. By the end of the academic year, an average increase was noted across all anthropometric traits when compared to earlier on in the year. Table 1 summarizes the aggregated anthropometric data at each time point. There was a significant increase in average body weight, by 1.55 kg (3.4 pounds; p<0.001), and in mean BMI, from 21.52 kg/m$^2$ to 22.16 kg/m$^2$ between the two time points (+0.64 kg/m$^2$, p<0.001). Notably, however, the mean BMI at both time points remained below 25 kg/m$^2$ indicating that the sample, on average, remained within the 'normal weight' category throughout the year. WC and HC also increased significantly, by 1.14 cm (p<0.001) and 0.93 cm (p<0.001) respectively. While a modest rise in WHR was noted between the two time points, it did not reach the threshold for statistical significance (P = 0.083). There was no significant difference in anthropometric change (i.e. change in weight, BMI, WC, HC, and WHR) between those who were followed for less than, or more than, the average follow-up time (21.6 weeks). In terms of their BMI categories, by the end of the academic year 77.1% (n = 189) of the participants were in the normal weight range, 8.6% were underweight (n = 21), 11.4% were overweight (n = 28), and 2.9% (n = 7) were obese.

## Sex-specific trends: Anthropometric presentation at baseline

Table 2 presents the sex-specific trends across all anthropometric traits at the beginning of the year (i.e. baseline). Overall, males presented with larger body weight (p<0.001), higher BMI (p = 0.008), larger WC (p<0.001), larger HC (p<0.001), and a higher WHR (p<0.001) at baseline, as compared to their females counterparts.

## Sex-specific trends: Anthropometric changes in first year of university

In terms of change from the beginning to the end of the year, both sexes saw an increase across all measured anthropometric characteristics. Separate subgroup analyses of anthropometric change from baseline, for both the males and females, revealed significant gains in both genders groups for body weight (males: p<0.001, females: p<0.001), BMI (males: p<0.001,

**Table 1. Overall trends in first year of university.**

|  | Beginning Mean (SD) | End Mean (SD) | Change MD (95% CI) | P-value* |
|---|---|---|---|---|
| **Body Weight (kg)** | 60.42 (11.98) | 61.97 (12.39) | 1.55 (1.24–1.86) | **<0.001** |
| **BMI (kg/m$^2$)** | 21.52 (3.34) | 22.16 (3.45) | 0.65 (0.53–0.76) | **<0.001** |
| **Waist Circumference (cm)** | 75.08 (8.69) | 76.27 (8.99) | 1.14 (0.63–1.66) | **<0.001** |
| **Hip Circumference (cm)** | 97.18 (7.73) | 98.11 (7.44) | 0.93 (0.55–1.31) | **<0.001** |
| **WHR** | 0.772 (0.050) | 0.776 (0.054) | 0.004 (-0.001–0.009) | 0.083 |

Data are expressed as mean (SD) and mean difference (95% CI); Abbreviations: BMI, body mass index; WHR, Waist to hip ratio; MD, Mean difference. *Pairwise comparison *via* Wilcoxon sign rank test (non-adjusted analysis of change in outcomes from the beginning to the end of the school year). P-values below 0.05 represented in bold font.

**Table 2. Baseline differences between Male (n = 48) and Female (n = 197) students at the beginning of the 1st year.**

| Anthropometric Trait | | Beginning of the Year Mean (SD) | P-value[*] |
|---|---|---|---|
| **Body Weight (kg)** | Males | 71.37 (12.68) | **<0.001** |
| | Females | 57.76 (10.18) | |
| **BMI (kg/m$^2$)** | Males | 22.62 (3.79) | **0.008** |
| | Females | 21.25 (3.18) | |
| **Waist Circumference (cm)** | Males | 81.38 (9.28) | **<0.001** |
| | Females | 73.55 (7.83) | |
| **Hip Circumference (cm)** | Males | 100.56 (7.57) | **<0.001** |
| | Females | 96.36 (7.56) | |
| **WHR** | Males | 0.808 (0.040) | **<0.001** |
| | Females | 0.763 (0.048) | |

All data presented as mean (SD); Abbreviations: BMI, body mass index; WHR, Waist to hip ratio. [*]Non-parametric comparison *via* Mann Whitney U test (non-adjusted comparison of males vs. females at baseline). P-values below 0.05 represented in bold font.

females: p<0.001), WC (males: p = 0.006, females: p<0.001), and HC (males: p = 0.006, females: p<0.001) from the beginning to the end of the year (S1 and S2 Tables). In comparison, no significant change in WHR was noted in both subgroups (males: p = 0.173, females: p = 0.193). Comparing the magnitude of change between the two sex groups showed that males gained slightly more body weight than females (1.90 kg *vs.*1.46 kg respectively). Overall, this trend was consistent across the other measured obesity traits as well, wherein males displayed moderately larger gains in BMI (0.74 kg/m2 vs. 0.62 kg/m2), WC (1.76cm vs. 0.99 cm), HC (1.08cm vs. 0.89cm), and WHR (0.0085 vs. 0.0030) towards the end of first year in university, compared to females. However, none of the observed differences in the magnitude of change between males and females were found to be statistically significant (interaction: p>0.05 across all traits). Table 3 summarizes the sex-specific anthropometric trends from the beginning and end of first year university.

**Table 3. Sex-specific trends from the beginning to the end of first year in male (n = 48) and female (n = 197) undergraduate students.**

| | | Beginning Mean (SD) | End Mean (SD) | Change MD (95% CI) | Time[*] p and η$^2$ | Sex[*] p and η$^2$ | Interaction[*] p and η$^2$ |
|---|---|---|---|---|---|---|---|
| **Body Weight[1] (kg)** | Males | 71.37 (12.68) | 73.27 (13.12) | 1.90 (1.13–2.68) | **<0.001**; 0.079 | **<0.001**; 0.233 | 0.270; 0.005 |
| | Females | 57.76 (10.18) | 59.22 (10.53) | 1.46 (1.12–1.80) | | | |
| **BMI[2] (kg/m$^2$)** | Males | 22.62 (3.79) | 23.36 (3.90) | 0.74 (0.48–1.00) | **<0.001**; 0.064 | **0.002**; 0.041 | 0.640; 0.001 |
| | Females | 21.25 (3.18) | 21.87 (3.28) | 0.62 (0.49–0.76) | | | |
| **Waist Circumference[1] (cm)** | Males | 81.38 (9.28) | 83.14 (9.83) | 1.76 (0.66–2.85) | **0.005**; 0.033 | **<0.001**; 0.147 | 0.638; 0.001 |
| | Females | 73.55 (7.83) | 74.59 (7.93) | 0.99 (0.41–1.58) | | | |
| **Hip Circumference (cm)** | Males | 100.56 (7.57) | 101.64 (7.19) | 1.08 (0.29–1.88) | **<0.001**; 0.054 | **<0.001**; 0.067 | 0.506; 0.002 |
| | Females | 96.36 (7.56) | 97.25 (7.26) | 0.89 (0.46–1.32) | | | |
| **WHR[1]** | Males | 0.808 (0.040) | 0.816 (0.052) | 0.0085 (-0.0027–0.0197) | 0.373; 0.003 | **<0.001**; 0.140 | 0.645; 0.001 |
| | Females | 0.763 (0.048) | 0.767 (0.049) | 0.0030 (-0.0024–0.0084) | | | |

Data are expressed as mean (SD) and mean difference (95% CI), WC data not collected for one female participant. [*] Significance from RMANOVA (Group: sex; Time: beginning to end); Rank based inverse normal transformation applied to all obesity traits; P-value threshold of 0.05 used for statistical significance; effect size determined by Partial Eta-Squared (η$^2$)

1. Body weight, WC, and WHR with living arrangement as a covariate, data on living arrangement was not collected for one participant

2. BMI with living arrangement and cohort of recruitment as covariates

## Discussion

In this investigation, we examined the effect of sex on obesity traits in first year of university. The investigation yielded several important results. Firstly, we found that males on average presented with larger body weight, BMI, WC, HC, and WHR at baseline as compared to females. Secondly, an overall net increase was observed in the sample, across all measured outcomes, towards the end of the academic year when compared to early on in the year. Notably, in this case, while significant gains in body weight, BMI, WC, and HC were noted, the change in WHR was not found to be significant. Thirdly, a consistent trend was observed in the two separate sex subgroups, wherein both males and females experienced significant growth in body weight, BMI, WC, and HC, but not WHR, during first year of university. Lastly, we found that while males displayed slightly larger gains than females over time, across all measured anthropometric parameters, the differences in the magnitude of change were modest with no significant sex effect being found within this context.

Weight gain in undergraduate students during first year of university has been extensively documented in previous studies from around the world [15,16]. Through our study, we confirmed this trend at McMaster University in Ontario, Canada. While the popular North American notion of 'Freshman 15' suggests that students gain approximately 15lbs (6.8kg) in first year of university, our results indicate a more modest overall increase of about 3.4lbs (1.55kg) on average. Nevertheless, this represents a significant change when compared to the general Canadian population. A report from Statistics Canada, involving data collected through the Canadian National Health Survey, previously indicated an average weight gain of 0.5 to 1 kg over a two-year period among Canadian adults [39]. Hence, in comparison, an average weight gain of 1.55 kg over a 5-month period among first-year university students represents a noteworthy change.

Our result is in line with the pooled weight gain estimates of 1.36 kg and 1.75kg, determined via previous meta-analyses by Vadeboncoeur *et al.*, and Vella-Zarb and Elgar respectively [15,16]. Particularly, in their subgroup investigation of Canadian studies, Vadeboncoeur *et al.* further reported a pooled weight gain estimate of 1.71kg for Canadian first year university students, which is also consistent with our finding [15]. However, in their study, Vadeboncoeur *et al.* detected significant heterogeneity ($I^2 = 86.5\%$) [15]. This is particularly interesting because the reported estimates of overall weight gain in Canadian reports, that include both males and females, have varied from 0.79kg to 1.5kg [15]. With respect to BMI, the increasing trend that we found is also consistent with what has been previously reported by Mifsud *et al.* and Beaudry *et al.* [29,30]. It is noteworthy that most studies within this context have primarily investigated only two anthropometric traits (i.e. body weight and BMI), with only a few examining additional parameters such as WC, HC, and WHR. With respect to WC, while our data indicates a significant increase over the academic year, consistent with the findings of Mifsud *et al.*, this result differs from that of Beaudry *et al.* which indicates no significant overall change in WC over time [29,30]. A similar inconsistency is noted between our result for WHR, which indicates no significant change over time, and that of Beaudry *et al.*, which indicates a significant rise over the academic year [29]. Overall, the observed heterogeneity in these findings can be partly attributed to the differences in either demographic factors (e.g. differences in ethnic distribution, baseline BMI distribution, sex ratios, living arrangement, academic program), and environmental factors (e.g. differences in campus environments and resources available on campus) across the different universities in Canada, or variation in methodological factors across studies (e.g. differences in sampling strategies, measurement strategies).

When examining differences between males and females, our results reveal sexual dimorphism of obesity traits at baseline, with males displaying significantly higher body weight,

BMI, WC, HC, and WHR than females. While most Canadian studies within this context have reported consistent sex specific trends at baseline, most prominently with respect to body weight and BMI, formal statistical testing of these baseline differences has been limited [29,30,40,41]. The sexual dimorphism of obesity traits has been extensively studied and can be attributed to fundamental biological differences between men and women across various age windows, such as differences in skeletal size, bone mass density, hormonal activity, and adipose tissue deposition [42–44].

With respect to change in obesity traits, our results indicate that both males and females experience significant gains across all measured obesity traits in first year of university. This indicates that both males and females are susceptible to gains in body weight and adiposity in first year of university. This trend is consistent with what has been previously reported for both male and female Canadian students within this context [15]. However, most notably, when comparing the average magnitude of change between males and females, we found no significant difference for any of the measured outcomes. Our finding aligns with that of a pooled subgroup analysis of 14 studies, by Vadeboncoeur *et al.*, which also indicates no difference in the amount of weight gained between males and female students [15]. Nevertheless, previous Canadian studies within this context have reported mixed results. While the findings reported by Pliner and Saunders, and Vella-Zarb and Elgar indicate no sex-based differences with respect to change in BMI and body weight respectively, studies by Mifsud *et al.*, and Beaudry *et al.* have evidenced significant sex-specific trends for change in body weight, BMI, WC, and WHR, but not HC [29–32]. Interestingly, all the aforementioned studies have been conducted in Ontario, Canada [29–32]. There are several possible reasons for the observed heterogeneity in findings, ranging from differences in population characteristics and campus environments at each university, to differences in study methodology. For instance, the ethnic distribution in our sample differs considerably from the aforementioned studies. Particularly, the samples in the studies by Mifsud *et al.*, Vella-Zarb RA *et al.*, and Beaudry *et al* are predominantly white Caucasian (>50%), with the latter two including only a modest proportion of Asian and African-Canadian students [29,30,32]. In comparison, our study sample presents a relatively diverse distribution, with the majority of the students being from the East Asian ethnic group, followed by considerable proportion of students from white Caucasian, South Asian, and Middle Eastern ethnic groups. Such factors may have an impact on the sex-specific trajectories of weight gain. Additionally, the baseline distribution of BMI weight status at a university can also impact the trajectory of BMI change. Hence, within this context, our differing results highlight the importance of conducting multiple studies not only across Canada but also within each province because multiple factors may differentiate university campuses from each other. Ultimately, a systematic review and meta-analysis of more studies, with exploration of between-study heterogeneity, will provide conclusive answers on the sexual dimorphism in change of obesity traits in first year, and its associated predictors in the Canadian undergraduate student population.

Our follow up investigation has several strengths, including a longitudinal study design, use of the same anthropometric parameters as the most recent study by Beaudry *et al*, investigation of participants from the same geographic region (i.e. Ontario), and use of the same statistical methodology. Furthermore, given that most Canadian studies so far have primarily examined either body weight or BMI outcomes within this context, our study provides valuable data on additional obesity parameters including WC, HC, and WHR, which is lacking in current literature. Limitations of our study include a modest sample size (N = 245) which is insufficiently powered to detect small effects. Additionally, we did not investigate physical activity as a covariate in our models, as done by Beaudry *et al*. in their study, due to a change in our method of measurement after the two first waves of recruitment. Similarly, we could not investigate

ethnicity as a potential covariate in our present analysis due to the limited sample size of certain ethnic subgroups in our overall study sample. Apart from that, we did not examine body composition parameters in our study and hence could not specifically characterize the observed anthropometric change. It is important to note here that the weight gain observed in our sample may not be entirely attributed to an increase in fat, but also to additional contributing factors such as continued development and/or increased physical activity, and increase in muscle mass. Unfortunately, in this case, we could not evaluate parameters such as lean mass or fat mass. Nonetheless, our investigation of adiposity indicators, such as waist and hip circumference, revealed significant increases in those areas among both male and female participants. Hence, based on our data, we postulate that one of the components contributing to the observed weight gain in our sample may possibly be a potential increase in fat. However, we acknowledge that there may be additional contributing factors as discussed above, and recognize that the data is limited in terms of the information it provides to characterize the observed change. In our study, we also witnessed a higher attrition rate than Beaudry *et al.* which may have potentially biased our study results. We are aware that using normal transformation is a question of debate in the statistical field [45]. Lastly, we acknowledge that our sample had a significant imbalance in the ratio of male to female participants (approximately 20:80). This imbalance in the sex ratio, along with insufficient power for detection of small effects, may have prevented us from detecting subtle sex differences in anthropometric change. However, it is important to note that most previous studies within this context have included a disproportionately larger percentage of female participants [15]. Furthermore, previous Canadian reports have shown varied results and our study results are at least consistent with some of those previous reports. Nevertheless, we recognize this is an important limitation that restricts our ability to make inferences with the results of this investigation. Lastly, as highlighted previously, in this investigation, we followed the analysis protocol outlined by Beaudry at al. to optimize our ability to compare our results. However, we are aware that alternative statistical methods can be utilized to analyze this data.

In conclusion, our data confirm significant weight gain in both male and female first year university students in Ontario, Canada. However, our data do not indicate sex specific differences within this context. Ultimately, our study contributes to current evidence on this unresolved topic and highlights the need of further studies in this area. It further highlights the importance of accounting for sex and gender in health research to make the findings more applicable to the population. Given the association of obesity with higher morbidity and mortality, understanding the predictors of weight gain in young adulthood may be critical in optimizing the prediction, prevention and treatment of obesity. Future studies may also consider investigating the mechanisms of weight gain in the undergraduate student population by sex/gender, through quantitative and qualitative approaches.

## Supporting information

**S1 Data.**
(XLSX)

**S1 Table. Distribution of demographic characteristics in the overall sample (n = 245) and in the male (n = 48) and female (n = 197) subgroups.**
(DOCX)

**S2 Table. Sex-specific trends in obesity traits from the beginning to the end of first year by male (n = 48) and female (n = 197) subgroups.**
(DOCX)

## Acknowledgments

We are indebted to all participants of this study. We would also like to extend our thanks to Anika Shah, Roshan Ahmad, Baanu Manoharan, Adrian Santhakumar, Kelly Zhu, Guneet Sandhu, Dea Sulaj, Tina Khordehi, Ansha Suleman, Heba Shahaed, Andrew Ng, Tania Mani, Sriyathavan Srichandramohan, Deven Deonarain, Celine Keomany, Omaike Sikder, Isis Lunsky, Gurudutt Kamath, and Christy Yu for their contribution.

## Author Contributions

**Conceptualization:** David Meyre.

**Data curation:** Tanmay Sharma.

**Formal analysis:** Tanmay Sharma, David Meyre.

**Funding acquisition:** David Meyre.

**Investigation:** Tanmay Sharma, Rita E. Morassut, Christine Langlois, David Meyre.

**Methodology:** David Meyre.

**Project administration:** David Meyre.

**Resources:** David Meyre.

**Supervision:** David Meyre.

**Validation:** David Meyre.

**Writing – original draft:** Tanmay Sharma, David Meyre.

**Writing – review & editing:** Rita E. Morassut, Christine Langlois.

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
