## [Decision Letter · Decision Letter 0]

25 Sep 2020

PONE-D-20-20612

Effect of sex/gender on obesity traits in Canadian first year university students: the GENEiUS study

PLOS ONE

Dear Dr. Meyre,

Thank you for submitting your manuscript to PLOS ONE. After careful consideration, we feel that it has merit but does not fully meet PLOS ONE’s publication criteria as it currently stands. Therefore, we invite you to submit a revised version of the manuscript that addresses the points raised during the review process.

We look forward to receiving your revised manuscript.

Kind regards,

Mauro Lombardo

Academic Editor

PLOS ONE

Journal Requirements:

Reviewers' comments:

Reviewer's Responses to Questions

**Comments to the Author**

1. Is the manuscript technically sound, and do the data support the conclusions?

Reviewer #1: Yes

Reviewer #2: Partly

Reviewer #3: Yes

Reviewer #4: Yes

Reviewer #5: Yes

2. Has the statistical analysis been performed appropriately and rigorously? 

Reviewer #1: Yes

Reviewer #2: Yes

Reviewer #3: Yes

Reviewer #4: Yes

Reviewer #5: I Don't Know

3. Have the authors made all data underlying the findings in their manuscript fully available?

Reviewer #1: Yes

Reviewer #2: Yes

Reviewer #3: Yes

Reviewer #4: Yes

Reviewer #5: No

4. Is the manuscript presented in an intelligible fashion and written in standard English?

Reviewer #1: Yes

Reviewer #2: Yes

Reviewer #3: Yes

Reviewer #4: Yes

Reviewer #5: Yes

5. Review Comments to the Author

Reviewer #1: This is a prospective observational cohort study that investigated the sex-specific changes in obesity features during first year of university. Two hundred and forty five first year undergraduate students were followed over 1 year with data collected early in the academic year and towards the end of the year. Sex-stratified analysis showed significant increase in weight, BMI, and waist and hip circumferences in both males and females. However, a comparison of the magnitude of changes over time between the two sex groups did not reach a threshold for statistical significance for any of the investigated traits.

Manuscript is well-written, with logical sequence and easy to understand. Furthermore, it is timely and relevant to current obesity problem.

I have no major issues with the methods and results or the conclusion the authors draw from them. Some minor comments are as follows:

1- Method: Study is well designed and method is appropriate. However, there is too much explanation about statistical analysis. This section can be written more concise. Instead, previous paper like Beaudry et al, (2019) can be cited.

2- Results: Results are clear and well organized. However, some data in the text (lines 195-203) are duplicated in table 1. Authors can choose to present them in either text or the table.

Reviewer #2: Thanks for the opportunity to review the article entitled "Effect of sex/gender on obesity traits in Canadian first year university students: the GENEiUS study". Although I found the topic interesting, the investigation is limited within the scope of the hypothesis formulated and tested.

Comments:

Demographics: I do not agree with the claim that the study recruited a "multi-ethnic" sample. The categories used to group participants by race and ethnicity might not capture their actual cultural and ethnic backgrounds.Also, your study would benefit from a table that includes the key demographic characteristics in your sample.

Moderator / grouping variable: Sex was a key study indicator in your study. Yet, females (n=197) outnumbered males (n=48) 4 times over. Your acknowledgement is valid, but it still makes it difficult to make inferences grouping the data in such a way (pg. 17, ln 346-349).

Small-to-no significant effect sizes: I am wondering if your investigation could in a new direction after learning from failure to reject your null hypothesis. I am optimistic that such a finding is a meaningful finding, which perhaps leads to sub-group analyses or studies with more depth. For example, how about collecting information on the processes that might be shaping weight gain by sex/gender? What are some differences and similarities that might explain each group's weight gain? A qualitative study might be a key next step.

Reviewer #3: The manuscript of Sharma et al. describes the weight gain that occurs during the first year of college in students at McMaster University in Ontario (Canada). The average gain was around 1.5 kg, with no significant differences between the sexes.

It is a well-written and interesting paper.

Only one aspect that I consider important to include in the manuscript. In the paper, the weight gain that occurs during the period studied seems to be a consequence of the university stay (change of environment and habits, sedentary lifestyle ...). However, it is not clear whether this weight change also occurs in the general population of similar age. A global significant weight increase has been described during adulthood. Therefore, this Reviewer thinks that it would be important to compare the data obtained in this study with those that can be found in general population. It would be interesting to have some group of non-university people available to compare, but since this is not the case, it is at least necessary to make the comparison with data available in Ontario from general population.

Since it is possible that part of the differences found between studies could be caused by the ethnic distribution of the students, I do not know if it would be possible to subdivide, at least the group of women, more numerous, thus comparing if there are differences associated with ethnic distribution.

Reviewer #4: Sharma et al. compose a very well written manuscript that longitudinally examines the 'freshman 15' in a group of undergraduate students in Ontario Canada, with an emphasis on sex-specific differences in weight gain. After review of the manuscript, it is clear that the authors considered their data prior to interpretation and do a great job highlighting both the strengths and weaknesses with the discussion of this manuscript. As mentioned by the authors a major weakness is the major discrepancy in males vs. females. While this is a limiting factor that had the potential to limit their ability to detect significant differences, the authors demonstrate that others have found similar results suggesting there may be generalizability to their findings. Overall I thought the manuscript was technically sound and well-written. I only have a few minor concerns.

1. Throughout the manuscript, the authors dive into the idea of obesity and the potential for the 'freshman 15' to contribute to obesity development. While this is likely true, the authors only briefly in the discussion touch on the possibility for continued development. This would be particularly true for males as many can undergo late-stage development during their early college years. Therefore, its unclear if the increase in weight gain for males is truly a reflection of an obesogenic environment or development.

2. In addition, without the knowledge of physical activity among these individuals, we again can not be sure if the weight gain is purely fat mass or a combination of fat mass and muscle mass (particularly if PA was increased). Again this is not likely, but should be more thoughtfully developed in the discussion.

3. Lastly, this group largely falls within the realm of normal weight status with only a small percentage falling in the category of overweight and an even smaller percent falling in the category of obese. This is important to note as it may not reflect other Universities, for instance those in the south in the USA where obesity rates can be over 40%. Again, are these students obesogenic?

Reviewer #5: In this work, Sharma and colleagues perform a longitudinal study investigating the morphometric evolution of first year University students in the McMaster University (Ontario, Canada). During this first year, several previous reports had shown a significant increase in BW and BMI. Some of them had reported significant sex-specific trends in these parameters, although the existing literature was heterogeneous and not conclusive. This new study, although including a low number of individuals (245), confirms a significant increase in BW, BMI, WC and HC in the students, with no significant difference between sexes, although males showed a tendency to greater increases in BW, BMI, WC and HC. This work, although limited, adds to previous reports using a longitudinal and relatively more complete assessment of morphometric parameters, and could be of use for future meta-analysis.

I have several comments:

1.- Statistical methods: it is not clear why authors choose to use the non-parametric tests for baseline and pairwise comparison of outcomes (Tables 1 and 2); while using the RMANOVA test for other year- and sex-specific outcomes. I suggest that outcomes are analyzed for their parametric or non-parametric distribution, justifying and homogenizing the choice of the statistical tests used.

2.- Authors claim several times throughout the text as a relevant finding that males presented with larger BW, WC or HC than females, while these parameters are usually larger in males than in females. In my opinion, only the higher BMI and WHR in males is a relevant finding, showing a stronger tendency to obesity in males.

3.- Since the mean topic of the work is the difference of sex in weight gain during first-year university, the ratio male/female in the cohort is a very important piece of information. However, it is not shown until the Results section. Here, authors report that 80.4% of the sample were female, but it is not clear to what cohort is this % referring to. Since it is presented in the same paragraph where the initial recruited population is described, (n=361), I understand that it 80.4% of this population are females. Therefore, only 76 males were included in the study. However, in Table 2 authors indicate a 19.6% (n=48) males, so maybe the 80.4% females indicated earlier in the Results section referred to the population that completed the follow-up. This piece of information (% of females/males at every stage of the study) is very important for this precise work but is not clearly shown in the text until the final part of the Discussion section. I think that it should be emphasized and made clear from the beginning, including the abstract.

Also, and related to this, the ethnicity and living arrangements were only shown for the total population, but not indicated for each sex, which could also shed some light to the differences.

4.- Tables 1 and 2: indicate the precise statistical methods used for these tables. I assume, from the Methods section, that the Wilcoxon signed-rank test was performed for Table 1, and the Mann-Whitney U test was used for Table 2, since Table 1 shows the paired data and Table 2 shows the baseline. But I think it should be stated in the legends for clarity.

5.- In Table 3 there are a few unclear notes (1, 2), that are hard to find in the table text. In particular, it is not clear to me why only BW, WC and WHR were calculated using living arrangement as a covariate, and BMI also used cohort as covariate. Also, it is not clear to me the nature of the “cohort” covariate, a clear explanation would be welcome.

Also it would be interesting to see if ethnicity was a relevant covariate. I assume that, since it was not included, it was not; but I would mention it in the text. Author indicate ethnicity as a possible reason for the discrepancy between this study and previous ones; but the data presented in this study does not indicate that ethnicity is a relevant covariate.

6.- Authors mention in the Discussion section an increase in BMI with no increase in WHR; a deeper discussion of the nutritional and metabolic meaning of this finding would be welcome.

7.- Full data are not available in a public repository, and there is no Data availability statement in the text. Please, provide a link to download the full, anonymized data in a public repository.

In all, I think that this study is well performed and adds robust data to the existing studies that could be of use to the specialized public. I consider this work worthy of publication in PLOS-ONE, once the previous comments are addressed.

6. PLOS authors have the option to publish the peer review history of their article (what does this mean?). If published, this will include your full peer review and any attached files.

Reviewer #1: **Yes: **Baran Hashemi

Reviewer #2: No

Reviewer #3: No

Reviewer #4: No

Reviewer #5: No

---

## [Author Response · Author response to Decision Letter 0]

9 Nov 2020

We would like to thank the editor and the reviewers for their exceptional input and suggestions on the article. We have addressed their comments to the best of our ability and we think that the revised version of the manuscript has significantly improved. 

Reviewer #1

1- Method: Study is well designed and method is appropriate. However, there is too much explanation about statistical analysis. This section can be written more concise. Instead, previous paper like Beaudry et al, (2019) can be cited.

We thank the reviewer for this comment. We acknowledge the reviewer’s perspective regarding the statistical methods being slightly explanation-heavy for some readers, and hence have made an effort to shorten the section slightly. However, most of the details included in that section have been retained as we feel they are necessary for a few different reasons. Firstly, the submission guidelines of PLOS One particularly require detailed reporting of statistical methods as to allow for replication of analysis by anyone who may be interested. As such, many of the details included in this section are based on the reporting criteria specified by the journal. Secondly, in this case, while we followed the methodology of Beaudry et al (2019) given the follow-up nature of our investigation, we further included a couple of additional elements of analysis in our paper (e.g. analyzing differences in obesity traits between males and females at baseline) that were not present in the previous paper by Beaudry and colleagues. Hence, our methods include a couple of additional aspects that are not present in the previous paper and thus require additional explanation. Lastly, while the reviewer is correct in mentioning that we can cite the paper by Beaudry et al. for all the overlapping methodologies between the two papers, we feel that this could affect the readability of our paper. Generally, when methods are simply referenced without inclusion of any details in the paper, it requires an extra effort on the part of the readers to go through a separate referenced and can hence create extra steps for readers. Thus, while we did reference the paper by Beaudry et al. (2019) for readers who may be interested in reviewing that source, we also included some of the fundamental methodological details in our paper in order to provide clarity and sufficient information for the average reader to understand the paper without having to consult additional sources. 

2- Results: Results are clear and well organized. However, some data in the text (lines 195-203) are duplicated in table 1. Authors can choose to present them in either text or the table.

We thank the reviewer for the comment. We re-reviewed that section and feel that it is important to include some of the important details in the text even if those details are available in the table. In this case, we chose to only reiterate the information pertaining to the ‘change in the obesity traits.’ Given that the ‘change in traits’ is central focus of our paper and a critical point of discussion later in the paper, we think that mentioning those results in the text, in addition to the table, is important to make sure that readers can take note of that information and keep that important information in mind as they read the paper. 

Reviewer #2 

1. Demographics: I do not agree with the claim that the study recruited a "multi-ethnic" sample. The categories used to group participants by race and ethnicity might not capture their actual cultural and ethnic backgrounds. Also, your study would benefit from a table that includes the key demographic characteristics in your sample.

We thank the reviewer for the comment. We agree with the reviewer that the composition of our sample may not be reflective of a truly ‘multi-ethnic’ sample that incorporates all the diverse cultural and ethnic backgrounds. We have now removed the term multi-ethnic as a description of our sample. Additionally, we have included a supplementary table with the key demographic characteristics of the sample as per the reviewer’s suggestion. 

2. Moderator / grouping variable: Sex was a key study indicator in your study. Yet, females (n=197) outnumbered males (n=48) 4 times over. Your acknowledgement is valid, but it still makes it difficult to make inferences grouping the data in such a way (pg. 17, ln 346-349).

Thank you for the comment. We agree with the reviewer that this is a limitation of the investigation and as mentioned by the reviewer, we have acknowledged this in the discussion section of our paper. As we discussed in the paper, most studies in this area present a similar imbalance in ratio of males and females, including the previous study by Beaudry et al (2019) that we followed. We have now added an additional sentence in the discussion section to further indicate that this imbalance in ratio of males to females is an important limitation that should be considered when interpreting the results of our study. 

“Lastly, we acknowledge that our sample had a significant imbalance in the ratio of male to female participants (approximately 20:80). This imbalance in the sex ratio, along with insufficient power for detection of small effects, may have prevented us from detecting subtle sex differences in anthropometric change. However, it is important to note that most previous studies within this context have included a disproportionately larger percentage of female participants [15]. Furthermore, previous Canadian reports have shown varied results and our study results are at least consistent with some of those previous reports. Nevertheless, we recognize this is an important limitation that restricts our ability to make inferences with the results of this investigation.”

3. Small-to-no significant effect sizes: I am wondering if your investigation could in a new direction after learning from failure to reject your null hypothesis. I am optimistic that such a finding is a meaningful finding, which perhaps leads to sub-group analyses or studies with more depth. For example, how about collecting information on the processes that might be shaping weight gain by sex/gender? What are some differences and similarities that might explain each group's weight gain? A qualitative study might be a key next step.

Thank you for the comment. In this case, since we did not find a difference in anthropometric change between males and females, our investigation did not merit further investigation into the different mechanisms of weight gain by sex/gender. Nevertheless, while this is beyond the scope of our paper, we do agree that this is an interesting field of research. We have now added this point as a recommendation for future studies in the discussion section of our paper. 

“Future studies may also consider investigating the mechanisms of weight gain in the undergraduate student population by sex/gender, through quantitative and qualitative approaches.”

Reviewer #3 

1. In the paper, the weight gain that occurs during the period studied seems to be a consequence of the university stay (change of environment and habits, sedentary lifestyle ...). However, it is not clear whether this weight change also occurs in the general population of similar age. A global significant weight increase has been described during adulthood. Therefore, this Reviewer thinks that it would be important to compare the data obtained in this study with those that can be found in general population. It would be interesting to have some group of non-university people available to compare, but since this is not the case, it is at least necessary to make the comparison with data available in Ontario from general population.

The reviewer brings up an excellent point. While we were not able to specifically find data pertaining to non-university Canadians in the same age group, we have added a broader comparison to the general Canadian population based on data available from Statistics Canada.

“Nevertheless, this represents a significant change when compared to the general Canadian population. A report from Statistics Canada, involving data collected through the Canadian National Health Survey, previously indicated an average weight gain of 0.5 to 1 kg over a two-year period among Canadian adults [36]. Hence, in comparison, an average weight gain of 1.55 kg over a 5-month period among first-year university students represents a noteworthy change.” 

2. Since it is possible that part of the differences found between studies could be caused by the ethnic distribution of the students, I do not know if it would be possible to subdivide, at least the group of women, more numerous, thus comparing if there are differences associated with ethnic distribution.

Thank you for the great comment. We acknowledge that examining the differences associated with ethnic distribution would be an interesting area of analysis. However, given our limited sample size in this case, we feel that an analysis of ethnic subgroups within the sex/gender subgroups would be severely underpowered. We think this analysis would also not be adequately powered with male group as our sample includes participants from multiple ethnic groups with the distribution of participants across each of the ethnic subgroups being unequal. Hence, the number of people within each ethnicity x sex subgroup is minimal and hence such analysis would be severely underpowered. Nonetheless, we agree with the reviewer that this is an interested area of investigation, and while it may not be possible to include the suggested analysis in this report due to power and sample size limitations, we have another paper that is currently in revision at PLOS One that independently examines the effect of race/ethnicity on anthropometric traits in the same cohort. 

Reviewer #4

1. Throughout the manuscript, the authors dive into the idea of obesity and the potential for the 'freshman 15' to contribute to obesity development. While this is likely true, the authors only briefly in the discussion touch on the possibility for continued development. This would be particularly true for males as many can undergo late-stage development during their early college years. Therefore, its unclear if the increase in weight gain for males is truly a reflection of an obesogenic environment or development.

We thank the reviewer for this important comment. We have now added a paragraph in the discussion section of the paper that addresses these points. 

“It is important to note here that the weight gain observed in our sample may not be entirely attributed to an increase in fat, but also to additional contributing factors such as continued development and increase in muscle mass. Unfortunately, in this case, we could not evaluate parameters such as lean mass or fat mass. Nonetheless, our investigation of adiposity indicators, such as waist and hip circumference, revealed significant increases in those areas among both male and female participants. Hence, based on our data, we postulate that one of the components contributing to the observed weight gain in our sample may possibly be a potential increase in fat. However, we acknowledge that there may be additional contributing factors as discussed above, and recognize that the data is limited in terms of the information it provides to characterize the observed change.”

2. In addition, without the knowledge of physical activity among these individuals, we again can not be sure if the weight gain is purely fat mass or a combination of fat mass and muscle mass (particularly if PA was increased). Again this is not likely, but should be more thoughtfully developed in the discussion.

The reviewer brings up a fair point. We have now extended the discussion about the limitations pertaining to the characterization of weight change in our study, including the lack of information about physical activity. 

“Additionally, we did not investigate physical activity as a covariate in our models, as done by Beaudry et al. in their study, due to a change in our method of measurement after the two first waves of recruitment. Furthermore, we did not examine body composition parameters in our study and hence could not specifically characterize the observed anthropometric change. It is important to note here that the weight gain observed in our sample may not be entirely attributed to an increase in fat, but also to additional contributing factors such as continued development and / or increased physical activity, and increase in muscle mass. Unfortunately, in this case, we could not evaluate parameters such as lean mass or fat mass. Nonetheless, our investigation of adiposity indicators, such as waist and hip circumference, revealed significant increases in those areas among both male and female participants. Hence, based on our data, we postulate that one of the components contributing to the observed weight gain in our sample may possibly be potential increase in fat. However, we acknowledge that there may be additional contributing factors as discussed above, and recognize that the data is limited in terms of the information it provides to characterize the observed change.”

3. Lastly, this group largely falls within the realm of normal weight status with only a small percentage falling in the category of overweight and an even smaller percent falling in the category of obese. This is important to note as it may not reflect other Universities, for instance those in the south in the USA where obesity rates can be over 40%. Again, are these students obesogenic?

We agree with the reviewer that there are differences in university environments. We talked about campus heterogeneity as an important factor through the discussion section of our paper. We have further added a note about the differing distribution of BMI weight status across universities.

“Additionally, the baseline distribution of BMI weight status at a university can also impact the trajectory of BMI change. Hence, within this context, our differing results highlight the importance of conducting multiple studies not only across Canada but also within each province because multiple factors may differentiate university campuses from each other.”

Apart from that, we have also added a sentence in the results section to highlight the distribution of BMI weight statuses observed in our sample by the end of the academic year. 

 “In terms of their BMI categories, by the end of the academic year 77.1% (n = 189) of the participants were in the normal weight range, 8.6% were underweight (n = 21), 11.4% were overweight (n = 28), and 2.9% (n = 7) were obese.”

Reviewer #5

1.- Statistical methods: it is not clear why authors choose to use the non-parametric tests for baseline and pairwise comparison of outcomes (Tables 1 and 2); while using the RMANOVA test for other year- and sex-specific outcomes. I suggest that outcomes are analyzed for their parametric or non-parametric distribution, justifying and homogenizing the choice of the statistical tests used.

Thank you for the comment. In this case, given that our investigation was a follow-up to a previous report by Beaudry et al (2019), we tried to largely follow the same analytical methodology as the paper by Beaudry et al. (i.e. RMANVOA). Nevertheless, we further included a couple of additional elements of analysis in our paper (e.g. analyzing differences in obesity traits between males and females at baseline) that were not present in the previous paper by Beaudry and colleagues. Particularly, we used non-parametric tests for evaluation of traits without adjustment for covariates and used the RMANOVA test with transformation for analysis of traits with adjustment for covariates. We recognize that there are different analytical approaches that be used to analyze the data. In fact, we used a different approach involving regression analysis for our other papers on the effect of living arrangement and ethnicity on anthropometric that are currently published (Sharma et al., PLOS One 2020) or under revision. In this case, we feel that the respective tests used here are appropriate, and do not see the need to homogenize the approach. 

2.- Authors claim several times throughout the text as a relevant finding that males presented with larger BW, WC or HC than females, while these parameters are usually larger in males than in females. In my opinion, only the higher BMI and WHR in males is a relevant finding, showing a stronger tendency to obesity in males.

Thank you for the comment. While we agree with the reviewer that BMI and WHR are critical parameters and could be the focus of a paper, we feel that it is still important to discuss the other parameters in the text as well considering that the data for these traits have been presented in the paper. Given that we conducted this investigation as a follow-up to the previous paper by Beaudry et al, we assessed and discussed the same traits as the ones discussed by Beaudry and colleagues. Additionally, in recent times, there has generally been an increasing amount of the literature on the prognostic value of indicators such as WC and hence including these results may be of interest to the readers.

3.- Since the mean topic of the work is the difference of sex in weight gain during first-year university, the ratio male/female in the cohort is a very important piece of information. However, it is not shown until the Results section. Here, authors report that 80.4% of the sample were female, but it is not clear to what cohort is this % referring to. Since it is presented in the same paragraph where the initial recruited population is described, (n=361), I understand that it 80.4% of this population are females. Therefore, only 76 males were included in the study. However, in Table 2 authors indicate a 19.6% (n=48) males, so maybe the 80.4% females indicated earlier in the Results section referred to the population that completed the follow-up. This piece of information (% of females/males at every stage of the study) is very important for this precise work but is not clearly shown in the text until the final part of the Discussion section. I think that it should be emphasized and made clear from the beginning, including the abstract. Also, and related to this, the ethnicity and living arrangements were only shown for the total population, but not indicated for each sex, which could also shed some light to the differences.

This is an excellent point. We have now updated the text to better reflect the cohort information and have included a supplementary table the described the ethnicity and living arrangement distribution by sex/gender.

4.- Tables 1 and 2: indicate the precise statistical methods used for these tables. I assume, from the Methods section, that the Wilcoxon signed-rank test was performed for Table 1, and the Mann-Whitney U test was used for Table 2, since Table 1 shows the paired data and Table 2 shows the baseline. But I think it should be stated in the legends for clarity.

The reviewer brings up an important point. We have now updated the legends of both the tables as per the reviewer’s recommendation.

5.- In Table 3 there are a few unclear notes (1, 2), that are hard to find in the table text. In particular, it is not clear to me why only BW, WC and WHR were calculated using living arrangement as a covariate, and BMI also used cohort as covariate. Also, it is not clear to me the nature of the “cohort” covariate, a clear explanation would be welcome. Also it would be interesting to see if ethnicity was a relevant covariate. I assume that, since it was not included, it was not; but I would mention it in the text. Author indicate ethnicity as a possible reason for the discrepancy between this study and previous ones; but the data presented in this study does not indicate that ethnicity is a relevant covariate.

Thank you for the comment. In this case, we used the same statistical methodology and covariate adjustment strategy as the previous paper by Beaudry et al (2019). As such, in accordance with the protocol of Beadry et al, the covariates were only retained in the model if their interaction with the main effect (i.e. time) was significant or marginally significant. We have included this detail in the statistical methods section and have included a reference to the paper by Beaudry et al. (2019) for readers who are interested in reading their protocol in further detail. Cohort refers to the cohort of recruitment in this case. We have updated the legend of Table 3 to better reflect this. While we agree that ethnicity is an interesting variable to test, we did not explore it as a covariate in this case particularly because it was not also included as a covariate by Beaudry et al. in their paper. Hence, in order to keep the methodology consistent, we only explored the variables that were evaluated in the previous paper. However, we so have another paper that is currently in revision at PLOS One that examines the effect of ethnicity on anthropometric traits in GENEiUS. 

6.- Authors mention in the Discussion section an increase in BMI with no increase in WHR; a deeper discussion of the nutritional and metabolic meaning of this finding would be welcome.

We thank the reviewer for this inspired comment. We have now added a paragraph in the discussion section of the paper that addresses these points. 

“It is important to note here that the weight gain observed in our sample may not be entirely attributed to an increase in fat, but also to additional contributing factors such as continued development and increase in muscle mass. Unfortunately, in this case, we could not evaluate parameters such as lean mass or fat mass. Nonetheless, our investigation of adiposity indicators, such as waist and hip circumference, revealed significant increases in those areas among both male and female participants. Hence, based on our data, we postulate that one of the components contributing to the observed weight gain in our sample may possibly be a potential increase in fat. However, we acknowledge that there may be additional contributing factors as discussed above, and recognize that the data is limited in terms of the information it provides to characterize the observed change.”

7.- Full data are not available in a public repository, and there is no Data availability statement in the text. Please, provide a link to download the full, anonymized data in a public repository.

Thank you for bringing up this important point. The dataset has now been included.

---

## [Decision Letter · Decision Letter 1]

26 Nov 2020

PONE-D-20-20612R1

Effect of sex/gender on obesity traits in Canadian first year university students: the GENEiUS study

PLOS ONE

Dear Dr. Meyre,

Thank you for submitting your manuscript to PLOS ONE. After careful consideration, we feel that it has merit but does not fully meet PLOS ONE’s publication criteria as it currently stands. Therefore, we invite you to submit a revised version of the manuscript that addresses the points raised during the review process.

We look forward to receiving your revised manuscript.

Kind regards,

Mauro Lombardo

Academic Editor

PLOS ONE

Reviewers' comments:

Reviewer's Responses to Questions

**Comments to the Author**

1. If the authors have adequately addressed your comments raised in a previous round of review and you feel that this manuscript is now acceptable for publication, you may indicate that here to bypass the “Comments to the Author” section, enter your conflict of interest statement in the “Confidential to Editor” section, and submit your "Accept" recommendation.

Reviewer #3: All comments have been addressed

Reviewer #4: All comments have been addressed

Reviewer #5: (No Response)

2. Is the manuscript technically sound, and do the data support the conclusions?

Reviewer #3: Yes

Reviewer #4: Yes

Reviewer #5: Yes

3. Has the statistical analysis been performed appropriately and rigorously? 

Reviewer #3: Yes

Reviewer #4: Yes

Reviewer #5: I Don't Know

4. Have the authors made all data underlying the findings in their manuscript fully available?

Reviewer #3: Yes

Reviewer #4: Yes

Reviewer #5: Yes

5. Is the manuscript presented in an intelligible fashion and written in standard English?

Reviewer #3: Yes

Reviewer #4: Yes

Reviewer #5: Yes

6. Review Comments to the Author

Reviewer #3: (No Response)

Reviewer #4: Sharma et al. have re-submitted their revised manuscript. It appears that the comments have been addressed adequately and that the authors have updated their methods, results, data and discussion to reflect these changes. While all points were not addressed in the manuscript (i.e. methods revisions), the authors provide adequate rationale for this decision. In this case, the authors left the methods more detailed for appropriate reproducibility as opposed to referring the reader/reviewer to other manuscripts for methodology. This follows with PLoS One's guidelines.

Reviewer #5: 1.- Statistical methods: it is not clear why authors choose to use the non-parametric tests for baseline and pairwise comparison of outcomes (Tables 1 and 2); while using the RMANOVA test for other year- and sex-specific outcomes. I suggest that outcomes are analyzed for their parametric or non-parametric distribution, justifying and homogenizing the choice of the statistical tests used.

A: Thank you for the comment. In this case, given that our investigation was a follow-up to a previous report by Beaudry et al (2019), we tried to largely follow the same analytical methodology as the paper by Beaudry et al. (i.e. RMANVOA). Nevertheless, we further included a couple of additional elements of analysis in our paper (e.g. analyzing differences in obesity traits between males and females at baseline) that were not present in the previous paper by Beaudry and colleagues. Particularly, we used non- parametric tests for evaluation of traits without adjustment for covariates and used the RMANOVA test with transformation for analysis of traits with adjustment for covariates. We recognize that there are different analytical approaches that be used to analyze the data. In fact, we used a different approach involving regression analysis for our other papers on the effect of living arrangement and ethnicity on anthropometric that are currently published (Sharma et al., PLOS One 2020) or under revision. In this case, we feel that the respective tests used here are appropriate, and do not see the need to homogenize the approach.

R: it sounds odd that the only argument to justify the use of a statistical method is “that others also used it before”, rather than proving that the statistical method is the most appropriate for the kind of data analyzed.

2.- Authors claim several times throughout the text as a relevant finding that males presented with larger BW, WC or HC than females, while these parameters are usually larger in males than in females. In my opinion, only the higher BMI and WHR in males is a relevant finding, showing a stronger tendency to obesity in males.

A: Thank you for the comment. While we agree with the reviewer that BMI and WHR are critical parameters and could be the focus of a paper, we feel that it is still important to discuss the other parameters in the text as well considering that the data for these traits have been presented in the paper. Given that we conducted this investigation as a follow-up to the previous paper by Beaudry et al, we assessed and discussed the same traits as the ones discussed by Beaudry and colleagues. Additionally, in recent times, there has generally been an increasing amount of the literature on the prognostic value of indicators such as WC and hence including these results may be of interest to the readers.

R: I think my point was not well understood. I agree that analyzing changes in BW, WC and HC of individuals with time is a very valuable analysis. What I do not agree is on the treatment as a relevant finding of the difference between male and female volunteers in BW, WC and HC: males always display larger BW, WC and HC in average than females, that is not biologically relevant.

3.- Since the mean topic of the work is the difference of sex in weight gain during first-year university, the ratio male/female in the cohort is a very important piece of information. However, it is not shown until the Results section. Here, authors report that 80.4% of the sample were female, but it is not clear to what cohort is this % referring to. Since it is presented in the same paragraph where the initial recruited population is described, (n=361), I understand that it 80.4% of this population are females. Therefore, only 76 males were included in the study. However, in Table 2 authors indicate a 19.6% (n=48) males, so maybe the 80.4% females indicated earlier in the Results section referred to the population that completed the follow-up. This piece of information (% of females/males at every stage of the study) is very important for this precise work but is not clearly shown in the text until the final part of the Discussion section. I think that it should be emphasized and made clear from the beginning, including the abstract.

Also, and related to this, the ethnicity and living arrangements were only shown for the total population, but not indicated for each sex, which could also shed some light to the differences.

A: This is an excellent point. We have now updated the text to better reflect the cohort information and have included a supplementary table the described the ethnicity and living arrangement distribution by sex/gender.

R: thank you for your efforts. I still do not see any mention to the males/females ratio in the abstract, where I think that it is a very relevant piece of information. Please, indicate where exactly in the text these changes were included, to be able to verify them easily.

4.- Tables 1 and 2: indicate the precise statistical methods used for these tables. I assume, from the Methods section, that the Wilcoxon signed-rank test was performed for Table 1, and the Mann-Whitney U test was used for Table 2, since Table 1 shows the paired data and Table 2 shows the baseline. But I think it should be stated in the legends for clarity.

A: The reviewer brings up an important point. We have now updated the legends of both the tables as per the reviewer’s recommendation.

R: thank you for your efforts, it is more complete now.

5.- In Table 3 there are a few unclear notes (1, 2), that are hard to find in the table text. In particular, it is not clear to me why only BW, WC and WHR were calculated using living arrangement as a covariate, and BMI also used cohort as covariate. Also, it is not clear to me the nature of the “cohort” covariate, a clear explanation would be welcome.

Also it would be interesting to see if ethnicity was a relevant covariate. I assume that, since it was not included, it was not; but I would mention it in the text. Author indicate ethnicity as a possible reason for the discrepancy between this study and previous ones; but the data presented in this study does not indicate that ethnicity is a relevant covariate.

A: Thank you for the comment. In this case, we used the same statistical methodology and covariate adjustment strategy as the previous paper by Beaudry et al (2019). As such, in accordance with the protocol of Beadry et al, the covariates were only retained in the model if their interaction with the main effect (i.e. time) was significant or marginally significant. We have included this detail in the statistical methods section and have included a reference to the paper by Beaudry et al. (2019) for readers who are interested in reading their protocol in further detail. Cohort refers to the cohort of recruitment in this case. We have updated the legend of Table 3 to better reflect this. While we agree that ethnicity is an interesting variable to test, we did not explore it as a covariate in this case particularly because it was not also included as a covariate by Beaudry et al. in their paper. Hence, in order to keep the methodology consistent, we only explored the variables that were evaluated in the previous paper. However, we so have another paper that is currently in revision at PLOS One that examines the effect of ethnicity on anthropometric traits in GENEiUS.

R: Thank you for your efforts. I understand that only the covariates that interacted significantly (or marginally singifficantly) with the main effect (time) were included. It would be useful to know the exact degree of significance for the analyzed variables, since the term “marginally” significant is not very clear. Also, authors indicate that they did not include ethnicity “because it was not included in Beaudry et al”; again, this is not a valid answer, because this is a different study. Was ethnicity significantly interacting with time? If it was, it should be included; if notit was not, that is a valid argument, not Beaudry’s.

6.- Authors mention in the Discussion section an increase in BMI with no increase in WHR; a deeper discussion of the nutritional and metabolic meaning of this finding would be welcome.

A: We thank the reviewer for this inspired comment. We have now added a paragraph in the discussion section of the paper that addresses these points.

“It is important to note here that the weight gain observed in our sample may not be entirely attributed to an increase in fat, but also to additional contributing factors such as continued development and increase in muscle mass. Unfortunately, in this case, we could not evaluate parameters such as lean mass or fat mass. Nonetheless, our investigation of adiposity indicators, such as waist and hip circumference, revealed significant increases in those areas among both male and female participants. Hence, based on our data, we postulate that one of the components contributing to the observed weight gain in our sample may possibly be a potential increase in fat. However, we acknowledge that there may be additional contributing factors as discussed above, and recognize that the data is limited in terms of the information it provides to characterize the observed change.”

R: Thank you for the new discussion.

7.- Full data are not available in a public repository, and there is no Data availability statement in the text. Please, provide a link to download the full, anonymized data in a public repository.

A: Thank you for bringing up this important point. The dataset has now been included.

R: Thank you for this dataset, it is very useful.

7. PLOS authors have the option to publish the peer review history of their article (what does this mean?). If published, this will include your full peer review and any attached files.

Reviewer #3: No

Reviewer #4: No

Reviewer #5: No

---

## [Author Response · Author response to Decision Letter 1]

28 Dec 2020

We thank the editor and reviewers for their comments and feedback. We have addressed their comments below to the best of our ability. 

Reviewer #4: Sharma et al. have re-submitted their revised manuscript. It appears that the comments have been addressed adequately and that the authors have updated their methods, results, data and discussion to reflect these changes. While all points were not addressed in the manuscript (i.e. methods revisions), the authors provide adequate rationale for this decision. In this case, the authors left the methods more detailed for appropriate reproducibility as opposed to referring the reader/reviewer to other manuscripts for methodology. This follows with PLoS One's guidelines.

Thank you so much for the constructive comment.

Reviewer #5: 1.- Statistical methods: it is not clear why authors choose to use the non-parametric tests for baseline and pairwise comparison of outcomes (Tables 1 and 2); while using the RMANOVA test for other year- and sex-specific outcomes. I suggest that outcomes are analyzed for their parametric or non-parametric distribution, justifying and homogenizing the choice of the statistical tests used.

A: Thank you for the comment. In this case, given that our investigation was a follow-up to a previous report by Beaudry et al (2019), we tried to largely follow the same analytical methodology as the paper by Beaudry et al. (i.e. RMANVOA). Nevertheless, we further included a couple of additional elements of analysis in our paper (e.g. analyzing differences in obesity traits between males and females at baseline) that were not present in the previous paper by Beaudry and colleagues. Particularly, we used non- parametric tests for evaluation of traits without adjustment for covariates and used the RMANOVA test with transformation for analysis of traits with adjustment for covariates. We recognize that there are different analytical approaches that be used to analyze the data. In fact, we used a different approach involving regression analysis for our other papers on the effect of living arrangement and ethnicity on anthropometric that are currently published (Sharma et al., PLOS One 2020) or under revision. In this case, we feel that the respective tests used here are appropriate, and do not see the need to homogenize the approach.

R: it sounds odd that the only argument to justify the use of a statistical method is “that others also used it before”, rather than proving that the statistical method is the most appropriate for the kind of data analyzed.

Thank you for the comment. We agree “that others also used it before” would not be a sufficient argument to apply the same approach. However, it seems the reviewer overlooked the other arguments we provided to justify our approach. Firstly, we think that the statistical methods proposed by Beaudry et al. are adequate to answer the questions asked. Secondly, the manuscript by Beaudry et al. has been reviewed and validated by experts, as part of the peer-review process, so we are not the only ones to think that the statistical approach used by Beaudry et al. is adequate. Thirdly, given the follow-up nature of our investigation, using a drastically different statistical approach may have added heterogeneity and impaired our ability to compare our results to those of Beaudry et al. Fourthly, we have provided additional elements of analysis in our paper that were not present in Beaudry’s paper and hence those methods differ – particularly with respect to using non-parametric tests for a raw comparison of traits in the absence of covariates, and the using RMANOVA, as done by Beaudry et al. for the comparison of traits with adjustment for covariates. We believe these are adequate reasons to justify our approach. However, we understand well the reviewer’s concern and we have added a limitation in the discussion to reflect their important point. 

“In this investigation, we followed the analysis protocol outlined by Beaudry at al. to optimize our ability to compare our results. However, we are aware that alternative statistical methods can also be utilized to analyze this data.” 

2.- Authors claim several times throughout the text as a relevant finding that males presented with larger BW, WC or HC than females, while these parameters are usually larger in males than in females. In my opinion, only the higher BMI and WHR in males is a relevant finding, showing a stronger tendency to obesity in males.

A: Thank you for the comment. While we agree with the reviewer that BMI and WHR are critical parameters and could be the focus of a paper, we feel that it is still important to discuss the other parameters in the text as well considering that the data for these traits have been presented in the paper. Given that we conducted this investigation as a follow-up to the previous paper by Beaudry et al, we assessed and discussed the same traits as the ones discussed by Beaudry and colleagues. Additionally, in recent times, there has generally been an increasing amount of the literature on the prognostic value of indicators such as WC and hence including these results may be of interest to the readers.

R: I think my point was not well understood. I agree that analyzing changes in BW, WC and HC of individuals with time is a very valuable analysis. What I do not agree is on the treatment as a relevant finding of the difference between male and female volunteers in BW, WC and HC: males always display larger BW, WC and HC in average than females, that is not biologically relevant.

Thank you for the clarification. While we agree with the reviewer that the observation that males always display larger BW, WC and HC on average is a well-established observation in literature, we still feel it is important to include this finding in our report for a couple of reasons. Firstly, we do think that males being larger than females is a biologically important observation. Secondly, in any case, we believe that before discussing the ‘change’ in traits across groups, it is imperative that the baseline distribution of the traits are assessed and reported, as knowing the baseline distribution provides context to better interpret values of change. In this case, while this may seem redundant given the extensive amount of literature on sex-based anthropometric patterns, we believe it is still important that we confirm this finding in our sample and provide that baseline context for the readers before providing a discussion on the magnitude of change observed in these traits. Apart from that, it should be noted that we only discussed this observation very briefly in our result and discussion sections with only a few lines dedicated this observation in the entire manuscript. We are aware that this is a common observation and hence it is definitely not something we have discussed frequently or emphasized as a core point of discussion in our manuscript. In fact, in our brief discussion of this observation, we also particularly acknowledge that this is a well-established observation in different age groups in literature (lines 312-315). Our discussion on the change in traits definitely makes up the majority of our paper.

3.- Since the mean topic of the work is the difference of sex in weight gain during first-year university, the ratio male/female in the cohort is a very important piece of information. However, it is not shown until the Results section. Here, authors report that 80.4% of the sample were female, but it is not clear to what cohort is this % referring to. Since it is presented in the same paragraph where the initial recruited population is described, (n=361), I understand that it 80.4% of this population are females. Therefore, only 76 males were included in the study. However, in Table 2 authors indicate a 19.6% (n=48) males, so maybe the 80.4% females indicated earlier in the Results section referred to the population that completed the follow-up. This piece of information (% of females/males at every stage of the study) is very important for this precise work but is not clearly shown in the text until the final part of the Discussion section. I think that it should be emphasized and made clear from the beginning, including the abstract.

Also, and related to this, the ethnicity and living arrangements were only shown for the total population, but not indicated for each sex, which could also shed some light to the differences.

A: This is an excellent point. We have now updated the text to better reflect the cohort information and have included a supplementary table the described the ethnicity and living arrangement distribution by sex/gender.

R: thank you for your efforts. I still do not see any mention to the males/females ratio in the abstract, where I think that it is a very relevant piece of information. Please, indicate where exactly in the text these changes were included, to be able to verify them easily.

The size of the analyzed sample and the gender ratio has now been specified in the abstract (line 29), methods section (lines 120-122), and the results section (line 172, 175). Thank you.

4.- Tables 1 and 2: indicate the precise statistical methods used for these tables. I assume, from the Methods section, that the Wilcoxon signed-rank test was performed for Table 1, and the Mann-Whitney U test was used for Table 2, since Table 1 shows the paired data and Table 2 shows the baseline. But I think it should be stated in the legends for clarity.

A: The reviewer brings up an important point. We have now updated the legends of both the tables as per the reviewer’s recommendation.

R: thank you for your efforts, it is more complete now.

Thank you so much for the feedback.

5.- In Table 3 there are a few unclear notes (1, 2), that are hard to find in the table text. In particular, it is not clear to me why only BW, WC and WHR were calculated using living arrangement as a covariate, and BMI also used cohort as covariate. Also, it is not clear to me the nature of the “cohort” covariate, a clear explanation would be welcome.

Also it would be interesting to see if ethnicity was a relevant covariate. I assume that, since it was not included, it was not; but I would mention it in the text. Author indicate ethnicity as a possible reason for the discrepancy between this study and previous ones; but the data presented in this study does not indicate that ethnicity is a relevant covariate.

A: Thank you for the comment. In this case, we used the same statistical methodology and covariate adjustment strategy as the previous paper by Beaudry et al (2019). As such, in accordance with the protocol of Beadry et al, the covariates were only retained in the model if their interaction with the main effect (i.e. time) was significant or marginally significant. We have included this detail in the statistical methods section and have included a reference to the paper by Beaudry et al. (2019) for readers who are interested in reading their protocol in further detail. Cohort refers to the cohort of recruitment in this case. We have updated the legend of Table 3 to better reflect this. While we agree that ethnicity is an interesting variable to test, we did not explore it as a covariate in this case particularly because it was not also included as a covariate by Beaudry et al. in their paper. Hence, in order to keep the methodology consistent, we only explored the variables that were evaluated in the previous paper. However, we so have another paper that is currently in revision at PLOS One that examines the effect of ethnicity on anthropometric traits in GENEiUS.

R: Thank you for your efforts. I understand that only the covariates that interacted significantly (or marginally singifficantly) with the main effect (time) were included. It would be useful to know the exact degree of significance for the analyzed variables, since the term “marginally” significant is not very clear. Also, authors indicate that they did not include ethnicity “because it was not included in Beaudry et al”; again, this is not a valid answer, because this is a different study. Was ethnicity significantly interacting with time? If it was, it should be included; if notit was not, that is a valid argument, not Beaudry’s.

Thank you for the comment. We have now specified our definition of marginal p-value (i.e. p<0.1) in the methods section of the paper for further clarity. With regards to the analysis of ethnicity as a covariate, as discussed previously, we did not initially intend on exploring ethnicity as a covariate as we followed the analytical protocol, covariate adjustment strategy, outlined by Beaudry et al. (2019) in order to minimize the heterogeneity between the study methodologies for better comparability of results. While this is a “different study, we believe that this is still an important consideration as changing the analytical strategy can introduce methodological heterogeneity and consequently reduce comparability, which is not favorable for a follow up investigation like ours. Nevertheless, we acknowledge the reviewer’s point and recognize the merit in assessing ethnicity as a potential covariate. However, in this case, we are unable to assess ethnicity as a covariate in the current investigation as the sample sizes of some of the represented ethnic groups in our study are too small and hence insufficient for adequate statistical adjustment. While our study sample includes certain large homogenous ethnic groups (i.e. East Asian, white-Caucasian, and South Asian), as we have indicated in Supplementary Table 1, the other ethnic groups in our sample are not sufficiently represented in terms of sample size. For example, the Middle Eastern ethnic group contains only 17 participants (6 males, 11 females). Similarly, in our sample there are 14 participants that collectively belong to other ethnic groups (e.g. African, Latin American etc.) with insufficient representation of each ethnic group in our study sample. As such, given that the number of participants in each of the individual ethnic categories are limited, we believe that the conclusions drawn based on such low subgroup sizes would be skewed due to inadequate power. We have now added this as a limitation in the discussion section of the paper to highlight this comment.

“We could not investigate ethnicity as a potential covariate in our present analysis due to the limited sample size of certain ethnic subgroups in our overall study sample.” 

6.- Authors mention in the Discussion section an increase in BMI with no increase in WHR; a deeper discussion of the nutritional and metabolic meaning of this finding would be welcome.

A: We thank the reviewer for this inspired comment. We have now added a paragraph in the discussion section of the paper that addresses these points.

“It is important to note here that the weight gain observed in our sample may not be entirely attributed to an increase in fat, but also to additional contributing factors such as continued development and increase in muscle mass. Unfortunately, in this case, we could not evaluate parameters such as lean mass or fat mass. Nonetheless, our investigation of adiposity indicators, such as waist and hip circumference, revealed significant increases in those areas among both male and female participants. Hence, based on our data, we postulate that one of the components contributing to the observed weight gain in our sample may possibly be a potential increase in fat. However, we acknowledge that there may be additional contributing factors as discussed above, and recognize that the data is limited in terms of the information it provides to characterize the observed change.”

R: Thank you for the new discussion.

Thank you so much for the feedback.

7.- Full data are not available in a public repository, and there is no Data availability statement in the text. Please, provide a link to download the full, anonymized data in a public repository.

A: Thank you for bringing up this important point. The dataset has now been included.

R: Thank you for this dataset, it is very useful.

Thank you so much for the feedback.

---

## [Decision Letter · Decision Letter 2]

12 Jan 2021

PONE-D-20-20612R2

Effect of sex/gender on obesity traits in Canadian first year university students: the GENEiUS study

PLOS ONE

Dear Dr. Meyre,

Thank you for submitting your manuscript to PLOS ONE. After careful consideration, we feel that it has merit but does not fully meet PLOS ONE’s publication criteria as it currently stands. Therefore, we invite you to submit a revised version of the manuscript that addresses the points raised during the review process.

We look forward to receiving your revised manuscript.

Kind regards,

Mauro Lombardo

Academic Editor

PLOS ONE

Reviewers' comments:

Reviewer's Responses to Questions

**Comments to the Author**

1. If the authors have adequately addressed your comments raised in a previous round of review and you feel that this manuscript is now acceptable for publication, you may indicate that here to bypass the “Comments to the Author” section, enter your conflict of interest statement in the “Confidential to Editor” section, and submit your "Accept" recommendation.

Reviewer #5: (No Response)

2. Is the manuscript technically sound, and do the data support the conclusions?

Reviewer #5: Yes

3. Has the statistical analysis been performed appropriately and rigorously? 

Reviewer #5: I Don't Know

4. Have the authors made all data underlying the findings in their manuscript fully available?

Reviewer #5: Yes

5. Is the manuscript presented in an intelligible fashion and written in standard English?

Reviewer #5: Yes

6. Review Comments to the Author

Reviewer #5: In this work, Sharma and colleagues perform a longitudinal study investigating the morphometric evolution of first year University students in the McMaster University (Ontario, Canada). During this first year, several previous reports had shown a significant increase in BW and BMI. Some of them had reported significant sex-specific trends in these parameters, although the existing literature was heterogeneous and not conclusive. This new study, although including a low number of individuals (245), confirms a significant increase in BW, BMI, WC and HC in the students, with no significant difference between sexes, although males showed a tendency to greater increases in BW, BMI, WC and HC. This work, although limited, adds to previous reports using a longitudinal and relatively more complete assessment of morphometric parameters, and could be of use for future meta-analysis.

I have several comments:

1.- Statistical methods: it is not clear why authors choose to use the non-parametric tests for baseline and pairwise comparison of outcomes (Tables 1 and 2); while using the RMANOVA test for other year- and sex-specific outcomes. I suggest that outcomes are analyzed for their parametric or non-parametric distribution, justifying and homogenizing the choice of the statistical tests used.

A: Thank you for the comment. In this case, given that our investigation was a follow-up to a previous report by Beaudry et al (2019), we tried to largely follow the same analytical methodology as the paper by Beaudry et al. (i.e. RMANVOA). Nevertheless, we further included a couple of additional elements of analysis in our paper (e.g. analyzing differences in obesity traits between males and females at baseline) that were not present in the previous paper by Beaudry and colleagues. Particularly, we used non- parametric tests for evaluation of traits without adjustment for covariates and used the RMANOVA test with transformation for analysis of traits with adjustment for covariates. We recognize that there are different analytical approaches that be used to analyze the data. In fact, we used a different approach involving regression analysis for our other papers on the effect of living arrangement and ethnicity on anthropometric that are currently published (Sharma et al., PLOS One 2020) or under revision. In this case, we feel that the respective tests used here are appropriate, and do not see the need to homogenize the approach.

R: it sounds odd that the only argument to justify the use of a statistical method is “that others also used it before”, rather than proving that the statistical method is the most appropriate for the kind of data analyzed.

A2: Thank you for the comment. We agree “that others also used it before” would not be a sufficient argument to apply the same approach. However, it seems the reviewer overlooked the other arguments we provided to justify our approach. Firstly, we think that the statistical methods proposed by Beaudry et al. are adequate to answer the questions asked. Secondly, the manuscript by Beaudry et al. has been reviewed and validated by experts, as part of the peer-review process, so we are not the only ones to think that the statistical approach used by Beaudry et al. is adequate. Thirdly, given the follow-up nature of our investigation, using a drastically different statistical approach may have added heterogeneity and impaired our ability to compare our results to those of Beaudry et al. Fourthly, we have provided additional elements of analysis in our paper that were not present in Beaudry’s paper and hence those methods differ –particularly with respect to using non-parametric tests for a raw comparison of traits in the absence of covariates, and the using RMANOVA, as done by Beaudry et al. for the comparison of traits with adjustment for covariates. We believe these are adequate reasons to justify our approach. However, we understand well the reviewer’s concern and we have added a limitation in the discussion to reflect their important point. “In this investigation, we followed the analysis protocol outlined by Beaudry at al. to optimize our ability to compare our results. However, we are aware that alternative statistical methods can also be utilized to analyze this data.

R2: I leave this matter for the Editor to decide. I still think that an evaluation of the parametric and non-parametric type of data would support the correct use of the chosen statistical methods, and would complete the information presented in the paper. Although I also acknowledge the reasons given by the authors for not changing the statistical methods from Beaudry et al., still if the data are not suited for a precise statistical method, using it is incorrect and could help fixing a previous mistake.

2.- Authors claim several times throughout the text as a relevant finding that males presented with larger BW, WC or HC than females, while these parameters are usually larger in males than in females. In my opinion, only the higher BMI and WHR in males is a relevant finding, showing a stronger tendency to obesity in males.

A: Thank you for the comment. While we agree with the reviewer that BMI and WHR are critical parameters and could be the focus of a paper, we feel that it is still important to discuss the other parameters in the text as well considering that the data for these traits have been presented in the paper. Given that we conducted this investigation as a follow-up to the previous paper by Beaudry et al, we assessed and discussed the same traits as the ones discussed by Beaudry and colleagues. Additionally, in recent times, there has generally been an increasing amount of the literature on the prognostic value of indicators such as WC and hence including these results may be of interest to the readers.

R: I think my point was not well understood. I agree that analyzing changes in BW, WC and HC of individuals with time is a very valuable analysis. What I do not agree is on the treatment as a relevant finding of the difference between male and female volunteers in BW, WC and HC: males always display larger BW, WC and HC in average than females, that is not biologically relevant.

A2: Thank you for the clarification. While we agree with the reviewer that the observation that males always display larger BW, WC and HC on average is a well-established observation in literature, we still feel it is important to include this finding in our report for a couple of reasons. Firstly, we do think that males being larger than females is a biologically important observation. Secondly, in any case, we believe that before discussing the ‘change’ in traits across groups, it is imperative that the baseline distribution of the traits are assessed and reported, as knowing the baseline distribution provides context to better interpret values of change. In this case, while this may seem redundant given the extensive amount of literature on sex-based anthropometric patterns, we believe it is still important that we confirm this finding in our sample and provide that baseline context for the readers before providing a discussion on the magnitude of change observed in these traits. Apart from that, it should be noted that we only discussed this observation very briefly in our result and discussion sections with only a few lines dedicated this observation in the entire manuscript. We are aware that this is a common observation and hence it is definitely not something we have discussed frequently or emphasized as a core point of discussion in our manuscript. In fact, in our brief discussion of this observation, we also particularly acknowledge that this is a well-established observation in different age groups in literature (lines 312-315). Our discussion on the change in traits definitely makes up the majority of our paper.

R2: I agree with the author’s response, thank you.

3.- Since the mean topic of the work is the difference of sex in weight gain during first-year university, the ratio male/female in the cohort is a very important piece of information. However, it is not shown until the Results section. Here, authors report that 80.4% of the sample were female, but it is not clear to what cohort is this % referring to. Since it is presented in the same paragraph where the initial recruited population is described, (n=361), I understand that it 80.4% of this population are females. Therefore, only 76 males were included in the study. However, in Table 2 authors indicate a 19.6% (n=48) males, so maybe the 80.4% females indicated earlier in the Results section referred to the population that completed the follow-up. This piece of information (% of females/males at every stage of the study) is very important for this precise work but is not clearly shown in the text until the final part of the Discussion section. I think that it should be emphasized and made clear from the beginning, including the abstract.

Also, and related to this, the ethnicity and living arrangements were only shown for the total population, but not indicated for each sex, which could also shed some light to the differences.

A: This is an excellent point. We have now updated the text to better reflect the cohort information and have included a supplementary table the described the ethnicity and living arrangement distribution by sex/gender.

R: thank you for your efforts. I still do not see any mention to the males/females ratio in the abstract, where I think that it is a very relevant piece of information. Please, indicate where exactly in the text these changes were included, to be able to verify them easily.

A2: The size of the analyzed sample and the gender ratio has now been specified in the abstract (line 29), methods section (lines 120-122), and the results section (line 172,175). Thank you.

R2: Thank you for this clarification.

4.- Tables 1 and 2: indicate the precise statistical methods used for these tables. I assume, from the Methods section, that the Wilcoxon signed-rank test was performed for Table 1, and the Mann-Whitney U test was used for Table 2, since Table 1 shows the paired data and Table 2 shows the baseline. But I think it should be stated in the legends for clarity.

A: The reviewer brings up an important point. We have now updated the legends of both the tables as per the reviewer’s recommendation.

R: thank you for your efforts, it is more complete now.

5.- In Table 3 there are a few unclear notes (1, 2), that are hard to find in the table text. In particular, it is not clear to me why only BW, WC and WHR were calculated using living arrangement as a covariate, and BMI also used cohort as covariate. Also, it is not clear to me the nature of the “cohort” covariate, a clear explanation would be welcome.

Also it would be interesting to see if ethnicity was a relevant covariate. I assume that, since it was not included, it was not; but I would mention it in the text. Author indicate ethnicity as a possible reason for the discrepancy between this study and previous ones; but the data presented in this study does not indicate that ethnicity is a relevant covariate.

A: Thank you for the comment. In this case, we used the same statistical methodology and covariate adjustment strategy as the previous paper by Beaudry et al (2019). As such, in accordance with the protocol of Beadry et al, the covariates were only retained in the model if their interaction with the main effect (i.e. time) was significant or marginally significant. We have included this detail in the statistical methods section and have included a reference to the paper by Beaudry et al. (2019) for readers who are interested in reading their protocol in further detail. Cohort refers to the cohort of recruitment in this case. We have updated the legend of Table 3 to better reflect this. While we agree that ethnicity is an interesting variable to test, we did not explore it as a covariate in this case particularly because it was not also included as a covariate by Beaudry et al. in their paper. Hence, in order to keep the methodology consistent, we only explored the variables that were evaluated in the previous paper. However, we so have another paper that is currently in revision at PLOS One that examines the effect of ethnicity on anthropometric traits in GENEiUS.

R: Thank you for your efforts. I understand that only the covariates that interacted significantly (or marginally singifficantly) with the main effect (time) were included. It would be useful to know the exact degree of significance for the analyzed variables, since the term “marginally” significant is not very clear. Also, authors indicate that they did not include ethnicity “because it was not included in Beaudry et al”; again, this is not a valid answer, because this is a different study. Was ethnicity significantly interacting with time? If it was, it should be included; if notit was not, that is a valid argument, not Beaudry’s.

A2: Thank you for the comment. We have now specified our definition of marginal p-value (i.e. p<0.1) in the methods section of the paper for further clarity. With regards to the analysis of ethnicity as a covariate, as discussed previously, we did not initially intend on exploring ethnicity as a covariate as we followed the analytical protocol, covariate adjustment strategy, outlined by Beaudry et al. (2019) in order to minimize the heterogeneity between the study methodologies for better comparability of results. While this is a “different study, we believe that this is still an important consideration as changing the analytical strategy can introduce methodological heterogeneity and consequently reduce comparability, which is not favorable for a follow up investigation like ours. Nevertheless, we acknowledge the reviewer’s point and recognize the merit in assessing ethnicity as a potential covariate. However, in this case, we are unable to assess ethnicity as a covariate in the current investigation as the sample sizes of some of the represented ethnic groups in our study are too small and hence insufficient for adequate statistical adjustment. While our study sample includes certain large homogenous ethnic groups (i.e. East Asian, white-Caucasian, and South Asian), as we have indicated in Supplementary Table 1, the other ethnic groups in our sample are not sufficiently represented in terms of sample size. For example, the Middle Eastern ethnic group contains only 17 participants (6 males, 11 females). Similarly, in our sample there are 14 participants that collectively belong to other ethnic groups (e.g. African, Latin American etc.) with insufficient representation of each ethnic group in our study sample. As such, given that the number of participants in each of the individual ethnic categories are limited, we believe that the conclusions drawn based on such low subgroup sizes would be skewed due to inadequate power. We have now added this as a limitation in the discussion section of the paper to highlight this comment. “We could not investigate ethnicity as a potential covariate in our present analysis due to the limited sample size of certain ethnic subgroups in our overall study sample.”

R2: Thank you for this new information, I think it is clearer now.

6.- Authors mention in the Discussion section an increase in BMI with no increase in WHR; a deeper discussion of the nutritional and metabolic meaning of this finding would be welcome.

A: We thank the reviewer for this inspired comment. We have now added a paragraph in the discussion section of the paper that addresses these points.

“It is important to note here that the weight gain observed in our sample may not be entirely attributed to an increase in fat, but also to additional contributing factors such as continued development and increase in muscle mass. Unfortunately, in this case, we could not evaluate parameters such as lean mass or fat mass. Nonetheless, our investigation of adiposity indicators, such as waist and hip circumference, revealed significant increases in those areas among both male and female participants. Hence, based on our data, we postulate that one of the components contributing to the observed weight gain in our sample may possibly be a potential increase in fat. However, we acknowledge that there may be additional contributing factors as discussed above, and recognize that the data is limited in terms of the information it provides to characterize the observed change.”

R: Thank you for the new discussion.

7.- Full data are not available in a public repository, and there is no Data availability statement in the text. Please, provide a link to download the full, anonymized data in a public repository.

A: Thank you for bringing up this important point. The dataset has now been included.

R: Thank you for this dataset, it is very useful.

7. PLOS authors have the option to publish the peer review history of their article (what does this mean?). If published, this will include your full peer review and any attached files.

Reviewer #5: No

---

## [Author Response · Author response to Decision Letter 2]

18 Jan 2021

We would like to thank the editor and reviewer 5 for their feedback on the second revision of our article. We have addressed their comments below to the best of our ability, and we think that the manuscript is now mature for publication. 

Reviewer #5: In this work, Sharma and colleagues perform a longitudinal study investigating the morphometric evolution of first year University students in the McMaster University (Ontario, Canada). During this first year, several previous reports had shown a significant increase in BW and BMI. Some of them had reported significant sex-specific trends in these parameters, although the existing literature was heterogeneous and not conclusive. This new study, although including a low number of individuals (245), confirms a significant increase in BW, BMI, WC and HC in the students, with no significant difference between sexes, although males showed a tendency to greater increases in BW, BMI, WC and HC. This work, although limited, adds to previous reports using a longitudinal and relatively more complete assessment of morphometric parameters, and could be of use for future meta-analysis.

We would like to thank the reviewer for the constructive comment.

I have several comments:

1.- Statistical methods: it is not clear why authors choose to use the non-parametric tests for baseline and pairwise comparison of outcomes (Tables 1 and 2); while using the RMANOVA test for other year- and sex-specific outcomes. I suggest that outcomes are analyzed for their parametric or non-parametric distribution, justifying and homogenizing the choice of the statistical tests used.

A: Thank you for the comment. In this case, given that our investigation was a follow-up to a previous report by Beaudry et al (2019), we tried to largely follow the same analytical methodology as the paper by Beaudry et al. (i.e. RMANVOA). Nevertheless, we further included a couple of additional elements of analysis in our paper (e.g. analyzing differences in obesity traits between males and females at baseline) that were not present in the previous paper by Beaudry and colleagues. Particularly, we used non- parametric tests for evaluation of traits without adjustment for covariates and used the RMANOVA test with transformation for analysis of traits with adjustment for covariates. We recognize that there are different analytical approaches that be used to analyze the data. In fact, we used a different approach involving regression analysis for our other papers on the effect of living arrangement and ethnicity on anthropometric that are currently published (Sharma et al., PLOS One 2020) or under revision. In this case, we feel that the respective tests used here are appropriate, and do not see the need to homogenize the approach.

R: it sounds odd that the only argument to justify the use of a statistical method is “that others also used it before”, rather than proving that the statistical method is the most appropriate for the kind of data analyzed.

A2: Thank you for the comment. We agree “that others also used it before” would not be a sufficient argument to apply the same approach. However, it seems the reviewer overlooked the other arguments we provided to justify our approach. Firstly, we think that the statistical methods proposed by Beaudry et al. are adequate to answer the questions asked. Secondly, the manuscript by Beaudry et al. has been reviewed and validated by experts, as part of the peer-review process, so we are not the only ones to think that the statistical approach used by Beaudry et al. is adequate. Thirdly, given the follow-up nature of our investigation, using a drastically different statistical approach may have added heterogeneity and impaired our ability to compare our results to those of Beaudry et al. Fourthly, we have provided additional elements of analysis in our paper that were not present in Beaudry’s paper and hence those methods differ –particularly with respect to using non-parametric tests for a raw comparison of traits in the absence of covariates, and the using RMANOVA, as done by Beaudry et al. for the comparison of traits with adjustment for covariates. We believe these are adequate reasons to justify our approach. However, we understand well the reviewer’s concern and we have added a limitation in the discussion to reflect their important point. “In this investigation, we followed the analysis protocol outlined by Beaudry at al. to optimize our ability to compare our results. However, we are aware that alternative statistical methods can also be utilized to analyze this data.

R2: I leave this matter for the Editor to decide. I still think that an evaluation of the parametric and non-parametric type of data would support the correct use of the chosen statistical methods, and would complete the information presented in the paper. Although I also acknowledge the reasons given by the authors for not changing the statistical methods from Beaudry et al., still if the data are not suited for a precise statistical method, using it is incorrect and could help fixing a previous mistake.

Thank you for the feedback. Once again, we respectfully disagree with the reviewer that performing non-parametric analyses on untransformed longitudinal data adjusted for covariates is needed in our study, and that for different reasons: 

1) The statistical methods proposed by Beaudry et al. are adequate to answer the questions asked. Anthropometric data are skewed and depart systematically from normality. Performing RMANOVA on non-normal longitudinal data following its inverse normal rank transformation is the best analytical strategy to deal with the lack of normality of data. 

2) The previous publication by Beaudry et al. has been published in PLOS One and has been validated by an academic editor and at least two reviewers. This is a strong indication that the statistical approach used by Beaudry et al. and more recently by us is adequate. If the reviewer really thinks that the statistical design in the Beaudry et al. publication is inadequate, their first action may be to post a comment on the PLOS One website and ask for a formal correction of the article. We checked the Beaudry et al. publication on the website and since its publication 1.5 years ago, no comment has been posted regarding the Baudry’s publication. Knowing that the article has been viewed by 5,800 people, the absence of negative comment strongly indicates that the reader’s community is comfortable with the statistical methods used in the paper. 

3) Given the follow-up nature of our investigation, using a drastically different statistical approach may have added heterogeneity and impaired our ability to compare our results to those of Beaudry et al. 

4) We have provided additional elements of analysis in our paper that were not present in Beaudry’s paper and hence those methods differ –particularly with respect to using non-parametric tests for a raw comparison of traits in the absence of covariates. As we use both parametric an non-parametric statistical analyses in our study, we do not understand why the reviewer is not satisfied. 

5) The method used by Beaudry et al. (RMANOVA with transformed data) is by far the most frequently used method in literature when it comes to analyzing longitudinal series of non-normal data with adjustment for covariates. As the reviewer did not provide any guidance on the type of statistical tests he wanted us to use, we had to make an extensive literature search to find examples of non-parametric tests that allow the analysis of longitudinal series of untransformed non-normal data with adjustment for covariates. Unfortunately, we did not find any alternative non-parametric method that we apply to our study. This indicates that the reviewer’s request is very unconventional.

We feel that the reviewer’s repeated request is kind of odd and is very challenging to address in practice. This is the third time we provide a strong justification for our analytical design, and we do not understand why the reviewer is not more receptive to our arguments. We leave this matter for the Editor to decide, but we hope he is satisfied with our answers. We have added a limitation in the discussion to reflect the reviewer’s comment. “In this investigation, we followed the analysis protocol outlined by Beaudry at al. to optimize our ability to compare our results. However, we are aware that alternative statistical methods may also be utilized to analyze this data.

2.- Authors claim several times throughout the text as a relevant finding that males presented with larger BW, WC or HC than females, while these parameters are usually larger in males than in females. In my opinion, only the higher BMI and WHR in males is a relevant finding, showing a stronger tendency to obesity in males.

A: Thank you for the comment. While we agree with the reviewer that BMI and WHR are critical parameters and could be the focus of a paper, we feel that it is still important to discuss the other parameters in the text as well considering that the data for these traits have been presented in the paper. Given that we conducted this investigation as a follow-up to the previous paper by Beaudry et al, we assessed and discussed the same traits as the ones discussed by Beaudry and colleagues. Additionally, in recent times, there has generally been an increasing amount of the literature on the prognostic value of indicators such as WC and hence including these results may be of interest to the readers.

R: I think my point was not well understood. I agree that analyzing changes in BW, WC and HC of individuals with time is a very valuable analysis. What I do not agree is on the treatment as a relevant finding of the difference between male and female volunteers in BW, WC and HC: males always display larger BW, WC and HC in average than females, that is not biologically relevant.

A2: Thank you for the clarification. While we agree with the reviewer that the observation that males always display larger BW, WC and HC on average is a well-established observation in literature, we still feel it is important to include this finding in our report for a couple of reasons. Firstly, we do think that males being larger than females is a biologically important observation. Secondly, in any case, we believe that before discussing the ‘change’ in traits across groups, it is imperative that the baseline distribution of the traits are assessed and reported, as knowing the baseline distribution provides context to better interpret values of change. In this case, while this may seem redundant given the extensive amount of literature on sex-based anthropometric patterns, we believe it is still important that we confirm this finding in our sample and provide that baseline context for the readers before providing a discussion on the magnitude of change observed in these traits. Apart from that, it should be noted that we only discussed this observation very briefly in our result and discussion sections with only a few lines dedicated this observation in the entire manuscript. We are aware that this is a common observation and hence it is definitely not something we have discussed frequently or emphasized as a core point of discussion in our manuscript. In fact, in our brief discussion of this observation, we also particularly acknowledge that this is a well-established observation in different age groups in literature (lines 312-315). Our discussion on the change in traits definitely makes up the majority of our paper.

R2: I agree with the author’s response, thank you.

We would like to thank the reviewer for the constructive comment.

3.- Since the mean topic of the work is the difference of sex in weight gain during first-year university, the ratio male/female in the cohort is a very important piece of information. However, it is not shown until the Results section. Here, authors report that 80.4% of the sample were female, but it is not clear to what cohort is this % referring to. Since it is presented in the same paragraph where the initial recruited population is described, (n=361), I understand that it 80.4% of this population are females. Therefore, only 76 males were included in the study. However, in Table 2 authors indicate a 19.6% (n=48) males, so maybe the 80.4% females indicated earlier in the Results section referred to the population that completed the follow-up. This piece of information (% of females/males at every stage of the study) is very important for this precise work but is not clearly shown in the text until the final part of the Discussion section. I think that it should be emphasized and made clear from the beginning, including the abstract.

Also, and related to this, the ethnicity and living arrangements were only shown for the total population, but not indicated for each sex, which could also shed some light to the differences.

A: This is an excellent point. We have now updated the text to better reflect the cohort information and have included a supplementary table the described the ethnicity and living arrangement distribution by sex/gender.

R: thank you for your efforts. I still do not see any mention to the males/females ratio in the abstract, where I think that it is a very relevant piece of information. Please, indicate where exactly in the text these changes were included, to be able to verify them easily.

A2: The size of the analyzed sample and the gender ratio has now been specified in the abstract (line 29), methods section (lines 120-122), and the results section (line 172,175). Thank you.

R2: Thank you for this clarification.

We would like to thank the reviewer for the constructive comment.

4.- Tables 1 and 2: indicate the precise statistical methods used for these tables. I assume, from the Methods section, that the Wilcoxon signed-rank test was performed for Table 1, and the Mann-Whitney U test was used for Table 2, since Table 1 shows the paired data and Table 2 shows the baseline. But I think it should be stated in the legends for clarity.

A: The reviewer brings up an important point. We have now updated the legends of both the tables as per the reviewer’s recommendation.

R: thank you for your efforts, it is more complete now.

5.- In Table 3 there are a few unclear notes (1, 2), that are hard to find in the table text. In particular, it is not clear to me why only BW, WC and WHR were calculated using living arrangement as a covariate, and BMI also used cohort as covariate. Also, it is not clear to me the nature of the “cohort” covariate, a clear explanation would be welcome.

Also it would be interesting to see if ethnicity was a relevant covariate. I assume that, since it was not included, it was not; but I would mention it in the text. Author indicate ethnicity as a possible reason for the discrepancy between this study and previous ones; but the data presented in this study does not indicate that ethnicity is a relevant covariate.

A: Thank you for the comment. In this case, we used the same statistical methodology and covariate adjustment strategy as the previous paper by Beaudry et al (2019). As such, in accordance with the protocol of Beadry et al, the covariates were only retained in the model if their interaction with the main effect (i.e. time) was significant or marginally significant. We have included this detail in the statistical methods section and have included a reference to the paper by Beaudry et al. (2019) for readers who are interested in reading their protocol in further detail. Cohort refers to the cohort of recruitment in this case. We have updated the legend of Table 3 to better reflect this. While we agree that ethnicity is an interesting variable to test, we did not explore it as a covariate in this case particularly because it was not also included as a covariate by Beaudry et al. in their paper. Hence, in order to keep the methodology consistent, we only explored the variables that were evaluated in the previous paper. However, we so have another paper that is currently in revision at PLOS One that examines the effect of ethnicity on anthropometric traits in GENEiUS.

R: Thank you for your efforts. I understand that only the covariates that interacted significantly (or marginally singifficantly) with the main effect (time) were included. It would be useful to know the exact degree of significance for the analyzed variables, since the term “marginally” significant is not very clear. Also, authors indicate that they did not include ethnicity “because it was not included in Beaudry et al”; again, this is not a valid answer, because this is a different study. Was ethnicity significantly interacting with time? If it was, it should be included; if notit was not, that is a valid argument, not Beaudry’s.

A2: Thank you for the comment. We have now specified our definition of marginal p-value (i.e. p<0.1) in the methods section of the paper for further clarity. With regards to the analysis of ethnicity as a covariate, as discussed previously, we did not initially intend on exploring ethnicity as a covariate as we followed the analytical protocol, covariate adjustment strategy, outlined by Beaudry et al. (2019) in order to minimize the heterogeneity between the study methodologies for better comparability of results. While this is a “different study, we believe that this is still an important consideration as changing the analytical strategy can introduce methodological heterogeneity and consequently reduce comparability, which is not favorable for a follow up investigation like ours. Nevertheless, we acknowledge the reviewer’s point and recognize the merit in assessing ethnicity as a potential covariate. However, in this case, we are unable to assess ethnicity as a covariate in the current investigation as the sample sizes of some of the represented ethnic groups in our study are too small and hence insufficient for adequate statistical adjustment. While our study sample includes certain large homogenous ethnic groups (i.e. East Asian, white-Caucasian, and South Asian), as we have indicated in Supplementary Table 1, the other ethnic groups in our sample are not sufficiently represented in terms of sample size. For example, the Middle Eastern ethnic group contains only 17 participants (6 males, 11 females). Similarly, in our sample there are 14 participants that collectively belong to other ethnic groups (e.g. African, Latin American etc.) with insufficient representation of each ethnic group in our study sample. As such, given that the number of participants in each of the individual ethnic categories are limited, we believe that the conclusions drawn based on such low subgroup sizes would be skewed due to inadequate power. We have now added this as a limitation in the discussion section of the paper to highlight this comment. “We could not investigate ethnicity as a potential covariate in our present analysis due to the limited sample size of certain ethnic subgroups in our overall study sample.”

R2: Thank you for this new information, I think it is clearer now.

We would like to thank the reviewer for the constructive comment.

6.- Authors mention in the Discussion section an increase in BMI with no increase in WHR; a deeper discussion of the nutritional and metabolic meaning of this finding would be welcome.

A: We thank the reviewer for this inspired comment. We have now added a paragraph in the discussion section of the paper that addresses these points.

“It is important to note here that the weight gain observed in our sample may not be entirely attributed to an increase in fat, but also to additional contributing factors such as continued development and increase in muscle mass. Unfortunately, in this case, we could not evaluate parameters such as lean mass or fat mass. Nonetheless, our investigation of adiposity indicators, such as waist and hip circumference, revealed significant increases in those areas among both male and female participants. Hence, based on our data, we postulate that one of the components contributing to the observed weight gain in our sample may possibly be a potential increase in fat. However, we acknowledge that there may be additional contributing factors as discussed above, and recognize that the data is limited in terms of the information it provides to characterize the observed change.”

R: Thank you for the new discussion.

7.- Full data are not available in a public repository, and there is no Data availability statement in the text. Please, provide a link to download the full, anonymized data in a public repository.

A: Thank you for bringing up this important point. The dataset has now been included.

R: Thank you for this dataset, it is very useful.

---

## [Decision Letter · Decision Letter 3]

27 Jan 2021

PONE-D-20-20612R3

Effect of sex/gender on obesity traits in Canadian first year university students: the GENEiUS study

PLOS ONE

Dear Dr. Meyre,

Thank you for submitting your manuscript to PLOS ONE. After careful consideration, we feel that it has merit but does not fully meet PLOS ONE’s publication criteria as it currently stands. Therefore, we invite you to submit a revised version of the manuscript that addresses the points raised during the review process.

We look forward to receiving your revised manuscript.

Kind regards,

Mauro Lombardo

Academic Editor

PLOS ONE

Reviewers' comments:

Reviewer's Responses to Questions

**Comments to the Author**

1. If the authors have adequately addressed your comments raised in a previous round of review and you feel that this manuscript is now acceptable for publication, you may indicate that here to bypass the “Comments to the Author” section, enter your conflict of interest statement in the “Confidential to Editor” section, and submit your "Accept" recommendation.

Reviewer #5: (No Response)

2. Is the manuscript technically sound, and do the data support the conclusions?

Reviewer #5: Yes

3. Has the statistical analysis been performed appropriately and rigorously? 

Reviewer #5: I Don't Know

4. Have the authors made all data underlying the findings in their manuscript fully available?

Reviewer #5: Yes

5. Is the manuscript presented in an intelligible fashion and written in standard English?

Reviewer #5: Yes

6. Review Comments to the Author

Reviewer #5: There is only one question remaining, all the other comments were properly answered by authors. Congratulations to the authors for their work.

This is my original question:

1.- Statistical methods: it is not clear why authors choose to use the non-parametric tests for baseline and pairwise comparison of outcomes (Tables 1 and 2); while using the RMANOVA test for other year- and sex-specific outcomes. I suggest that outcomes are analyzed for their parametric or non-parametric distribution, justifying and homogenizing the choice of the statistical tests used.

I am afraid that there has been a lack of communication with this issue in subsequent answers. The data obtained in the repeated measures of the study are not normal, and this is why authors use non-parametric tests (Mann-Whitney or Wilcoxon signed-rank tests) to analyze these data. However, when analyzing differences in anthropometric outcomes over the year and the effect of sex on anthropometric change, authors perform an inverse normal rank transformation that converts their non-normal data into normal data. This transformed data is proposed to be now suitable for RMANOVA analysis.

Probably this strategy has been used in other reports previously, and I assume that, since transformed data are now normal, the test is correct. I was not aware of the use of this statistical strategy, and this is why I asked. I strongly suggest that authors cite previous similar reports to support this statistical strategy (they mention the existence of several reports, but refer to none in particular).

I understand that using non-parametric tests for the analysis of non-normal repeated measures would have been the best option. There are several of these tests available:

https://onlinelibrary.wiley.com/doi/abs/10.1002/sim.4780101210

https://www.jstatsoft.org/article/view/v050i12

https://www.jstatsoft.org/article/view/v064i09

Actually, normal transformation is a question of debate in the statistical field (https://www.ncbi.nlm.nih.gov/pmc/articles/PMC2921808/).

However, I also understand that, if this transformation is a common practice, and it is the strategy followed by the paper that is being used as a model by the authors (Beaudry et al.,), it makes sense to keep the transformation + RMANOVA for this precise analysis, since, strictly speaking, transformed data are normal.

I have not more to say in this point. I would have acknowledged this simple clarification (transformed data are normal, once transformed) when I first asked. I think we all have learnt with this revision!.

7. PLOS authors have the option to publish the peer review history of their article (what does this mean?). If published, this will include your full peer review and any attached files.

Reviewer #5: No

---

## [Author Response · Author response to Decision Letter 3]

29 Jan 2021

Reviewer #5: There is only one question remaining, all the other comments were properly answered by authors. Congratulations to the authors for their work.

We would like to thank the editor and the reviewer for their feedback on the third revision of our article. We have addressed their comments below to the best of our ability. 

This is my original question:

1.- Statistical methods: it is not clear why authors choose to use the non-parametric tests for baseline and pairwise comparison of outcomes (Tables 1 and 2); while using the RMANOVA test for other year- and sex-specific outcomes. I suggest that outcomes are analyzed for their parametric or non-parametric distribution, justifying and homogenizing the choice of the statistical tests used.

I am afraid that there has been a lack of communication with this issue in subsequent answers. The data obtained in the repeated measures of the study are not normal, and this is why authors use non-parametric tests (Mann-Whitney or Wilcoxon signed-rank tests) to analyze these data. However, when analyzing differences in anthropometric outcomes over the year and the effect of sex on anthropometric change, authors perform an inverse normal rank transformation that converts their non-normal data into normal data. This transformed data is proposed to be now suitable for RMANOVA analysis.

Probably this strategy has been used in other reports previously, and I assume that, since transformed data are now normal, the test is correct. I was not aware of the use of this statistical strategy, and this is why I asked. I strongly suggest that authors cite previous similar reports to support this statistical strategy (they mention the existence of several reports, but refer to none in particular).

We thank the reviewer for the clarification. We now mention in the Methods section that the inverse normal rank transformation resulted in the normality of the transformed data distribution. This means that our RMANOVA analysis of these transformed data is valid, as acknowledged by the reviewer. We also cite previous reports where the same transformation has been used.

I understand that using non-parametric tests for the analysis of non-normal repeated measures would have been the best option. There are several of these tests available:

https://onlinelibrary.wiley.com/doi/abs/10.1002/sim.4780101210

https://www.jstatsoft.org/article/view/v050i12

https://www.jstatsoft.org/article/view/v064i09

Actually, normal transformation is a question of debate in the statistical field (https://www.ncbi.nlm.nih.gov/pmc/articles/PMC2921808/).

However, I also understand that, if this transformation is a common practice, and it is the strategy followed by the paper that is being used as a model by the authors (Beaudry et al.,), it makes sense to keep the transformation + RMANOVA for this precise analysis, since, strictly speaking, transformed data are normal.

The reviewer makes a relevant point, thank you. We have now added as a limitation in the Discussion that normal transformation is a question of debate in the statistical field. 

I have not more to say in this point. I would have acknowledged this simple clarification (transformed data are normal, once transformed) when I first asked. I think we all have learnt with this revision!

Once again, we thank the reviewer for the very helpful clarification.

---

## [Decision Letter · Decision Letter 4]

2 Feb 2021

Effect of sex/gender on obesity traits in Canadian first year university students: the GENEiUS study

PONE-D-20-20612R4

Dear Dr. Meyre,

We’re pleased to inform you that your manuscript has been judged scientifically suitable for publication and will be formally accepted for publication once it meets all outstanding technical requirements.

Kind regards,

Mauro Lombardo

Academic Editor

PLOS ONE

Additional Editor Comments (optional):

Reviewers' comments:

Reviewer's Responses to Questions

**Comments to the Author**

1. If the authors have adequately addressed your comments raised in a previous round of review and you feel that this manuscript is now acceptable for publication, you may indicate that here to bypass the “Comments to the Author” section, enter your conflict of interest statement in the “Confidential to Editor” section, and submit your "Accept" recommendation.

Reviewer #5: All comments have been addressed

2. Is the manuscript technically sound, and do the data support the conclusions?

Reviewer #5: Yes

3. Has the statistical analysis been performed appropriately and rigorously? 

Reviewer #5: Yes

4. Have the authors made all data underlying the findings in their manuscript fully available?

Reviewer #5: Yes

5. Is the manuscript presented in an intelligible fashion and written in standard English?

Reviewer #5: Yes

6. Review Comments to the Author

Reviewer #5: (No Response)

7. PLOS authors have the option to publish the peer review history of their article (what does this mean?). If published, this will include your full peer review and any attached files.

Reviewer #5: No

---

## [Editor Report · Acceptance letter]

5 Feb 2021

PONE-D-20-20612R4 

Effect of sex/gender on obesity traits in Canadian first year university students: the GENEiUS study 

Dear Dr. Meyre:

I'm pleased to inform you that your manuscript has been deemed suitable for publication in PLOS ONE. Congratulations! Your manuscript is now with our production department. 

Kind regards, 

on behalf of

Dr. Mauro Lombardo 

Academic Editor

PLOS ONE